# Questionnaire-free machine-learning method to predict depressive symptoms among community-dwelling older adults

Sri Susanty[1,2], Herdiantri Sufriyana [3,4], Emily Chia-Yu Su [3,5,6]*, Yeu-Hui Chuang [1,7]*

**1** School of Nursing, College of Nursing, Taipei Medical University, Taipei, Taiwan, **2** Nursing Study Program, Faculty of Medicine, Universitas Halu Oleo, Kendari, Southeast Sulawesi, Indonesia, **3** Graduate Institute of Biomedical Informatics, College of Medical Science and Technology, Taipei Medical University, Taipei, Taiwan, **4** Department of Medical Physiology, Faculty of Medicine, Universitas Nahdlatul Ulama Surabaya, Surabaya, Indonesia, **5** Clinical Big Data Research Center, Taipei Medical University Hospital, Taipei, Taiwan, **6** Research Center for Artificial Intelligence in Medicine, Taipei Medical University, Taipei, Taiwan, **7** Center for Nursing and Healthcare Research in Clinical Practice Application, Wan Fang Hospital, Taipei Medical University, Taipei, Taiwan

* yeuhui@tmu.edu.tw (YHC); emilysu@tmu.edu.tw (ECYS)

## Abstract

The 15-item Geriatric Depression Scale (GDS-15) is widely used to screen for depressive symptoms among older populations. This study aimed to develop and validate a questionnaire-free, machine-learning model as an alternative triage test for the GDS-15 among community-dwelling older adults. The best models were the random forest (RF) and deep-insight visible neural network by internal validation, but both performances were undifferentiated by external validation. The AUROC of the RF model was 0.619 (95% CI 0.610 to 0.627) for the external validation set with a non-local ethnic group. Our triage test can allow healthcare professionals to preliminarily screen for depressive symptoms in older adults without using a questionnaire. If the model shows positive results, then the GDS-15 can be used for follow-up measures. This preliminary screening will save a lot of time and energy for healthcare providers and older adults, especially those persons who are illiterate.

**Data Availability Statement:** All relevant data files are publicly available from the Github database (https://github.com/herdiantrisufriyana/colab_dep).

## Introduction

Depressive symptoms in older adults are commonly unidentified and complicated by concurrent cognitive impairment [1]. To screen depressive symptoms in older adults, the Geriatric Depression Scale (GDS) is one of the most commonly used questionnaires. A recent systematic review and meta-analysis found that the 15-item version (GDS-15) is the most accurate compared to the shorter or longer versions [2]. By questionnaire-free variables, demographic and physical health data from routine visits can be utilized as an electronic health record (EHR) indicator to triage patients for a mental health follow-up by GDS-15. This utilization is possible because older adults with depressive symptoms may present with more physical complaints, implying a psychological change that caregivers might overlook [3]. However, the accuracy of utilizing such data for a triage test is still unclear.

**Funding:** This study was funded by: (1) the Postdoctoral Accompanies Research Project from the National Science and Technology Council (NSTC) in Taiwan (grant no.: NSTC111-2811-E-038-003-MY2) to Herdiantri Sufriyana; and (2) the Ministry of Science and Technology (MOST) in Taiwan (grant nos.: MOST110-2628-E-038-001 and MOST111-2628-E-038-001-MY2), and the Higher Education Sprout Project from the Ministry of Education (MOE) in Taiwan (grant no.: DP2-111-21121-01-A-05) to Emily Chia-Yu Su. The funders had no role in study design, data collection and analysis, decision to publish, or preparation of the manuscript.

**Competing interests:** The authors have declared that no competing interests exist.

## Motivation

Depression affects 264 million people globally [4]. Due to the different tools used for screening depression, the prevalence range is considerably wide [5]. Previous studies differently reported the prevalence rates of depression among community-dwelling older adults in Sweden (7%) [6], the United States (9.8%) [7], Nigeria (52.0%) [8], India (34.4%) [9], Singapore (13%) [10], Turkey (25.2%) [11], Japan (24%) [12], South Korea (72.2%) [13], and Malaysia (59.1%) [14]. Although a definitive one uses the Diagnostic and Statistical Manual of Mental Disorders V (DSM-V), the GDS-15 is quite reliable for diagnosis. Prevalences in Sweden were quite similar between those based on the GDS-15 (7%) and DSM-IV-TR/DSM-V (6.6%) [6].

In addition to feelings of sadness, helplessness, and pessimism, an older adult with this disorder may also experience a decrease in mood, loss of motivation, physical weakness, sleep disturbances, a feeling of hopelessness, a lack of help, and difficulty concentrating [15]. Depression in later life, if not promptly treated, can result in worse outcomes, e.g., a decreased quality of life [16], sleep disturbances [17], attempted suicide [18, 19], and even death [20]. Early identification of depressive symptoms is essential for early interventions.

## Previous works

Almost all existing predictive models of depressive symptoms include questionnaire-based predictors, e.g., the Patient Health Questionnaire (PHQ), the Edinburgh Postnatal Depression Scale (EPDS), and the GDS [21]. More-frequent identification of patients with depression (with an area under the receiver operating characteristic [ROC] curve [AUROC] of 0.700, 95% CI 0.629 to 0.771) still needed several questionnaire-based screening tools [22]. They were a part of the Self-Reported Quick Inventory of Depressive Symptomatology (QIDS-SR) and Hamilton Depression Rating Scale (HAM-D). A previous study developed an extended pre-dictD algorithm to predict major depression 12~24 months later based on the DSM-IV (AUROC 0.728, 95% CI 0.675 to 0.781; $n$ = 2670), but this also needs a subject to fill in the 12-Item Short Form (SF-12) for two of the predictors [23].

One study utilized a wearable device to predict GDS-15 and HAM-D results in older adults (AUROC 0.96, 95% CI 0.91 to 0.99; $n$ = 47); unfortunately, the sample size was small, and a wearable device might not be affordable for some older adults [24]. However, no previous study developed a questionnaire-free method to predict depressive symptoms based on standard screening questionnaires in community-dwelling older adults.

## Intuition

Later-life (aged 60+) depression is associated with several factors, and their assessments can utilize routine databases at the first visit of a subject to a healthcare facility. Some of these factors never change, i.e., age [25], gender [26–28], and past employment status (i.e., before 60 years old) [29–31]. A few of these factors rarely change, i.e., current employment status [31, 32], education [33, 34], religion [35, 36], marital status [37], living status [38–40], and lifestyle [41]. However, many of these factors can change on a monthly to yearly basis, i.e., health status [41–43], morbidities [28], hearing loss [44], and oral health and missing teeth [45]. A prediction model may utilize these factors to develop a triage test for the GDS-15 at any time. At the same time, this test can reduce the screening frequency of GDS-15 by restricting respondents to only those who test positive according to the prediction model. It should be a part of an EHR system with automatic run based on pre-existing, required information in EHR.

However, developing this model under a traditional approach, i.e., using a logistic regression (LR) algorithm, may be insufficient. In addition to LR, we also need other machine learning algorithms, which is a field of science concerned with how machines learn from data [46],

not limited to those based on statistical probability theory. Machine learning is a part of artificial intelligence that emulates human intellectual actions [47]. Their use is already pervasive in recognizing objects in images, transcribing speech to text, aligning internet content to user preferences, and selecting relevant search results [48]. Many fields in medicine have used this approach to predict medical outcomes, e.g., oncology [49], cardiology and critical care [50, 51], and obstetrics [52]. Machine learning provides a more extensive search space to find the most-accurate model using simple predictors, e.g., routine data in electronic medical records [49, 53]. This study aimed to develop and validate a questionnaire-free model to predict the GDS-15 among community-dwelling older adults by machine learning.

## Methods

### Study design

This study followed the guidelines for developing and reporting machine learning predictive models in biomedical research [54] (see S1 Table in S1 File) and the prediction model risk of bias assessment tool (PROBAST) [55] (see S2 Table in S1 File). The PROBAST development was according to transparent reporting of a multivariable prediction model for individual prognosis or diagnosis (TRIPOD) guidelines [56]. However, the PROBAST included recent findings on developing and validating multivariable prediction models, including one by any machine learning algorithm [57, 58]. For clinicians, we also provided a checklist to assess the suitability of our model for clinical settings [59] (see S3 Table in S1 File). A web application (https://predme.app/pre_gds15) is available as a prototype, but the future implementation should incorporate the application into an EHR system for automatic prediction based on pre-existing information. We utilized a dataset from our previous project investigating loneliness and depression in older adults. From June to September 2019, the previous project collected this dataset using a cross-sectional design from 15 community health centers (CHCs) in Kendari, Indonesia ($n$ = 1381). All patients aged 60 years or older with clear consciousness who visited the CHCs were enrolled. We applied a random sampling technique stratified by the CHCs. Trained assessors who collected the data were blind to the study outcome. Taipei Medical University (TMU) waived ethical clearance for this study. Both the TMU Joint Institutional Review Board (approval no.: N201905105) and the Ethical Research Committee in Universitas Halu Oleo (approval no.: 954/UN.29.20/PPM/2018) granted the original study an ethical clearance. Verbal consent was informed and obtained from the participant.

### Data source

The dataset consisted of 19 attributes which were 17 candidate-predictor variables, one grouping variable, and one outcome variable. The candidate-predictor variables were: 1) age (years); 2) gender (male/female); 3) religious beliefs (Christian/Hindu/Moslem); 4) educational attainment (illiterate/primary/secondary/high school/university/other); 5) marital status (single/ married/separated or divorced/widowed); 6) children (number of persons); 7) living status (alone/with a family member but no spouse/with a spouse only/with family member and spouse/other); 8) currently employed (no/yes); 9) previously employed (no/yes); 10) income (in Indonesian rupiah (IDR)); 11) duration of visiting the CHC (in the number of years of routine visits); 12) comorbidities (number of conditions); 13) health condition (very good/good/ fair/poor/very poor); 14) hearing problems (no/yes); 15) visual problems (no/yes); 16) oral status (very good/good/fair/poor/very poor); and 17) medication (number of prescribed drugs). We used ethnicity (Bugis-Makassar/Buton/Muna/Tolaki/non-local ethnicity) as a grouping variable for data partitioning in order to develop and validate our predictive models (see

"Model Validation"). The outcome variable was depressive symptoms (no/yes), as defined in the next section.

## Outcome definition

As the predicted outcome, depressive symptoms were assessed based on the GDS. There were 15 questions to obtain a score (which ranged from 0 to 15). Some items give a point if answered positively, while others give a point if answered negatively. If the score exceeds 5, the scale suggests a person has depressive symptoms [60]. A participant answered the questions with the assistance of a trained assessor. The GDS questionnaire is described in S4 Table in S1 File. The trained assessor assisted a participant in filling out the GDS questionnaire. The assessor was also blind to the predictor information. Predictor data were demographic data and routine physical health check results collected at the same time as that for GDS. Other health-care givers collected the data without knowing the assessment results of depressive symptoms. This blinding avoided outcome leakage. It was also carefully handled for all the analytical procedures, as described after each description of the relevant procedures (see S5 Table in S1 File). The event definition for this prediction task was depressive symptoms based on the GDS-15. However, to comply with the sample size requirement of the model development (see "Predictors"), we treated the outcome with a smaller sample size between positive and negative as the event. Under-diagnosis causes missed cases of depressive symptoms when screened using the GDS-15, which leads to failure to prevent major depressive disorders. Meanwhile, over-diagnosis causes an increased frequency of the use of the GDS-15 for each older adult, which may lead to further misclassification because the repetitive screening may cause response fatigue and rush, which lead to higher measurement error [61]. Nonetheless, the risk of under-diagnosis outweighs that of over-diagnosis.

## Data pre-processing

We binarized all categorical predictors into 0 or 1 for "no" or "yes" as to whether a category applied to a participant (Fig 1). All numerical predictors were standardized using the mean and standard deviation (SD) but capped at the 2.5% and 97.5% quantiles as the respective minimum and maximum values. This standardization resulted in a value range of approximately -1.96 to 1.96. Then, we applied normalization by shifting the central value (i.e., zero) to 0.5 and scaling the range by half; thus, the numerical predictors were within a range of 0 to 1. We only used the mean and SD calculated from data partitioned for model development. Standardization used these values for numerical predictors in any data partitions. Therefore, this pre-processing procedure is possible for future data. We checked for missing values in the dataset. The only missing value was that in visual problems for one participant ($n = 1/1381$, 0.072%). This value was missing completely at random since we got this information from routine physical health check data. We imputed the missing value using multiple imputations by the chain equation method after data transformation using only data in the same data partition. Randomly, the missing value was a part of the data partitioned for model development.

## Predictors

We only used data partitioned for model development to conduct predictor extraction, representation, and selection (Fig 1). For candidate predictors, the binarized predictors were extracted only for those without a perfect separation problem in which the predictor existed in only one of the outcomes. Perfect separation may occur because of a sampling error [54]. Although this may also occur in populations, including this kind of predictor may mislead predictive modeling to choose that predictor as the strong one for predicting outcomes. Of 40

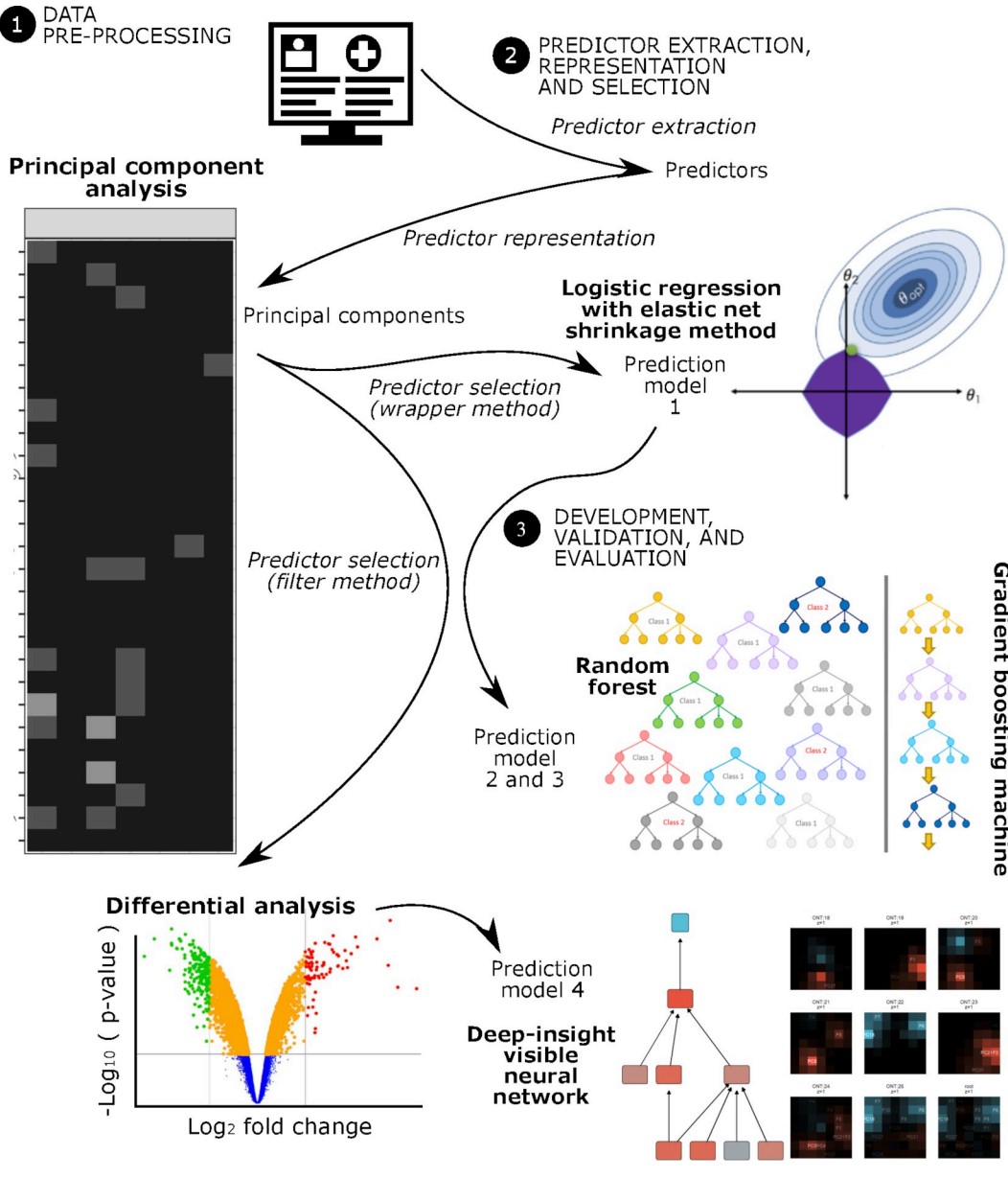

**Fig 1. Flowchart of the proposed methodology.**

predictors after binarization, only 37 were extracted. The excluded predictors were "other" living status, "very poor" oral status, and "Hindu" religion.

We assessed redundant predictors assisted by Pearson's correlation coefficients. Two binarized predictors were highly correlated ($r = 0.72$), which were "living with family members without a spouse" and a "widowed" marital status. We decided to retain these variables because the correlation was near borderline and was apparently due to sampling bias. A "widowed" marital status is not necessarily living with family members, while an older adult might live alone. Both predictors were not interchangeable.

To optimize the predictive performance, we applied a dimension-reduction technique using a principal component (PC) analysis (PCA) (Fig 1). We only used the top 19 PCs based

on the percent variance explained because we needed to comply with the sample size for predictive modeling based on PROBAST guidelines [58], which is 20 events per variable or candidate of predictors (see "Model Validation"). A ten-fold cross-validation procedure was applied on only data partitioned for model development. We used average values computed from ten rotated matrices of PCs to represent 37 binarized and numerical predictors into 19 PCs. We also used average values of data partitioned for model development to get those PCs for model validation. This study's resampled dimensional reduction method was already described elsewhere [62].

We also used other machine learning algorithms besides logistic regression to develop prediction models (see "Model Development"). However, the models required larger sample sizes of >50 events per variable [58]. We used the wrapper method in which we selected PCs using a logistic regression before being candidate predictors for the machine learning models (Fig 1). We applied the same hyperparameter tuning strategy of the LR for this predictor selection (see "Model Development").

## Model development

Although there are abundant machine learning algorithms for model development, we only partially compared the available algorithms (Fig 1). It is because a more models in comparison would be more vulnerable to a multiple-testing effect relative to the number of datasets, i.e., the best model is found simply by chance [63]. To avoid such comparison, we considered three criteria for choosing algorithms in developing the models: (1) those commonly used in clinical prediction studies, i.e., logistic regression [58], which expects a linear predictor-outcome correlation; (2) those which commonly outperformed others (177 algorithms) across 121 datasets [64], which allow a non-linear predictor-outcome correlation; and (3) our proposed neural-network algorithm [65], which pursues moderate predictive performance and deeper interpretability. A sufficient sample size was also considered according to the PROBAST guidelines since a small sample size was vulnerable to overfitting [58]. The three types of algorithms also covered those with the lowest and highest sample size requirements, which were 20 (i.e., logistic regression) and >200 (i.e., random forest [RF] and neural network) events per variable (EPVs), according to a previous study [66]. They also identified 50 and >200 EPVs for the decision tree and support vector machine. We did not use both, which neither commonly outperformed other algorithms nor required a sample size small enough for this study. Although we used algorithms that require >200 EPVs, we evaluated the models using rigorous data splitting. It would identify overfitting by comparing the evaluation results between internal and external validation sets. Both had the same and different characteristics for a particular circumstance (see Model validation), as recommended by the PROBAST guidelines [58].

In addition, we used a random-search method to tune values for the pre-defined hyperparameters. We also used those which were defined before conducting this study in a pre-registered protocol [65]. The randomness and pre-registration were deliberate to avoid a research bias, so-called "hypothesizing after the results are known (HARking)" [67]. In this study, HARking is a situation in which a set of hyperparameters for an algorithm, as a hypothesis, is preferably defined to achieve the only acceptable predictive performance in an external validation set.

We developed four models with different approaches. First, we applied the simplest model using logistic regression (LR) with a shrinkage method as recommended by the PROBAST guidelines (Fig 1). Instead of the PCs, this model used the 37 candidate predictors by an elastic net regression algorithm with L1- and L2-norm regularization. We chose this regularization

method over others to minimize the excluded predictors and prevent overfitting [58]. Hyperparameter tuning of this model used a random search with up to 10 configurations of alpha and lambda values as L1- and L2-norm regularization factors, respectively. We set the factors in the tradeoff between removing and maintaining the number of predictors used for predicting the outcome; thus, we could infer which variables have predictive values under a simple predictive modeling framework.

The second and third prediction models used RF and gradient boosting machine (GBM) algorithms (Fig 1). Both are state-of-the-art algorithms that consistently outperformed other algorithms across different outcomes [68]. The RF algorithm randomly selects some predictors to build multiple classification trees using subsets of samples in parallel. Meanwhile, the GBM sequentially applies a similar algorithm. Sequential application means a later tree in GBM is used to predict misclassification of earlier ones. Both algorithms are the most used competition-winning algorithms for predictions using tabular data compared to other 177 algorithms using 121 datasets [64]. While this is not outcome-specific, predictive modeling in a competition is independently validated; thus, the predictive performances of RF and GBM are considered reliable and reasonably evaluated. Hyperparameter tuning of these models also used a random search over six configurations of the number of predictors sampled at a time for RF and number of trees, maximum depth of a tree, and shrinkage factor for the GBM. Both models also configured for minimum samples per node. We defined these hyperparameter variables in aggregate between the tree-based ensemble learners to pursue a wide range of configurations. For example, we applied a different number of predictors sampled at a time for RF while maintaining the same tree structure. Contrarily, we applied different tree structures for GBM while maintaining the same number of predictors sampled a time. The best hyperparameters were selected for each of the algorithms under a variety of samples per node to take into account the effect of sampling error. Therefore, we expected a hypothesis search of hyperparameters well-covered while avoiding the pitfall of HARking.

The last prediction model used the deep-insight visible neural network (DI-VNN) algorithm (Fig 1). It is a deep-learning model or a convolutional neural network (CNN). This model emerged in recent years because it improves predictive performance for imaging data. The Deep Insight algorithm converts a non-image into image-like data as a multidimensional array in a meaningful way using a dimensional-reduction algorithm over the predictors. The VNN means that the network architecture is data-driven because it is determined based on a hierarchical clustering algorithm over the predictors. This approach addresses criticisms of the CNN as a black-box model, i.e., it is unexplained which features and how these result in a particular prediction; yet, a CNN model can predict an outcome very well. Details about the DI-VNN pipeline were previously described elsewhere [65]. Some modifications of this pipeline were those by applying this procedure over 37 predictors and 19 PCs, resulting in 18 candidate features for DI-VNN. These were centered using each average value after quantile-to-quantile normalization over all features among samples. To avoid HARking, we followed the same hyperparameter tuning approach, which was already pre-registered and thoroughly described elsewhere [65].

## Model validation

Data partitioning was conducted to obtain both internal and external validation sets. Respectively, this meant we had a training set and two test sets. We used participants with ethnicity not from Sulawesi Island for the external validation set. The model was expected for use in settings not limited to those with only the local ethnicities. Hence, we should test whether the model developed using data with local ethnicities would also have an acceptable predictive

performance if the model was applied to non-local ethnicities. This validation procedure may demonstrate the model's robustness in predicting outcomes in the general population [58].

We also randomly split the remaining set after excluding the external validation set. This procedure provided another external validation set with as much as ~20% of the remaining set. The first to third models applied 10-fold cross-validation and 30-time bootstrapping. Respectively, both were applied for hyperparameter tuning and model training with the best hyperparameters. We also applied 10-fold cross-validation to compute the rotated matrix of PCs. Meanwhile, the fourth model applied hold-out cross-validation with 80:20 ratios for the training and validation sets. To compare this model against the others, we applied 30-time bootstrapping to compute the predictive performance. To re-calibrate all models using logistic regression, we also applied 30-time bootstrapping.

### Evaluation metrics

We used the area under the receiver operating characteristics (ROC) curve (AUROC) as the primary evaluation metric. This selection was because the AUROC is threshold-agnostic. However, before evaluating this, we reported the calibration metric of a model using an LR in which the predicted probability as the model output became the only covariate. The models were well calibrated if the 95% CIs of the intercept and slope respectively covered 0 and 1, with the probability plots visually aligned with the reference line. Models were evaluated with and without re-calibration (see "Model Validation"). We chose all models that complied with the calibration metric. The best models were well-calibrated models that significantly outper-formed others, according to the AUROC. All metrics are reported with 95% CI. A model out-performed the others if the interval estimate was greater than the central value of the other models. Otherwise, more than one model might be selected. The best model was determined using the internal validation set. It also should be robust based on all external validation sets, for which the central value of the AUROC approximated >0.5, as a baseline value to determine if a predictive performance of a model was better than random or coin-flip guessing. Compared to the same baseline value, we also computed the specificity, accuracy, positive predictive value (PPV) or precision, and negative predictive value (NPV) using a threshold at approximately a sensitivity or recall of ~90% or a false negative rate of ~10% because the risk of under-diagnosis outweighs that of over-diagnosis. In addition, we explored the best model to identify important features post-analysis (see Results).

### Ontology analysis

Our DI-VNN model explore ontological relationships among the predictors in predicting the outcome. A detailed technical explanation of the DI-VNN algorithm was previously described elsewhere [65]. Briefly, there were three steps: (1) differential analysis for feature pre-selection; (2) structural representation of features; and (3) CNN model training.

We applied a differential analysis to choose 18 candidate features for DI-VNN among 37 predictors and 19 PCs (Fig 1). The differential analysis applied quantile-to-quantile normalization, which removed technical inter-variability (i.e., to measure predictors) across the subjects. By $t$-moderated statistics, a differential analysis selected candidate features (filter method for feature selection). The null hypothesis was that there is no significant difference in a feature value between positives and negatives. Since a predictor could be selected by chance, which posed the analysis to multiple testing bias, we adjusted the $p$-values using the Benjamini-Hoch-berg method. We selected a feature if the adjusted $p$-value or false discovery rate (FDR) was less than 0.05.

After pre-selection, the candidate features without the outcome were used to construct a structural representation of feature variabilities and inter-relationships. There were two types of structural representation (Fig 1): (1) spatial; and (2) hierarchical. We applied the *t*-distributed stochastic neighbor embedding (*t*-SNE) algorithm to cluster the selected features in a three-dimensional positioning spatially. A closer position means a higher correlation between a pair of features. Meanwhile, we applied clique-extracted ontology algorithm to cluster the selected features in a hierarchy. Features are more similar to those within the same ontology than those in a different ontology. Since these ontologies were hierarchical, we could evaluate which ontology was more predictive between that with fewer features (i.e., child ontology) and that with more features (i.e., parent ontology) after the model training.

Eventually, we used the representation as a CNN architecture and trained it using a backpropagation algorithm to predict the outcome. In CNN modeling, a maximum value would represent closer values in a multidimensional array. In this way, inter-relationships among features were also taken into account when predicting the outcome in addition to their values. The backpropagation algorithm in a CNN modeling also allowed us to signify which features and their inter-relationships were more weighted to predict the outcome. A more-extreme weight, either positively or negatively, was represented with a higher color intensity when visualizing the internal properties of our DI-VNN model. Therefore, using this ontology analysis, we could evaluate: (1) which set of features (i.e., ontology) were more predictive; (2) how these ontologies were connected; (3) what were important features in an ontology; and (4) how these features were related within an ontology.

## Results

### Most subjects have not obtained a university education, were not separated/divorced, and are religious believers

We developed four diagnostic prediction models using a cross-sectional dataset (*n* = 1252). These models were externally validated (*n* = 129) with non-local ethnic groups unobserved in the development sets (Table 1). Model validation may be challenging since estimates of depressive symptom prevalences in the validation set differed from those of the development sets. However, ethnicity may affect the distributions of the predictors and the outcome to some extent. A prediction model should be robust against the shift of data distribution (i.e., well-generalized). Therefore, our validation sets allowed the generalization test, including data with non-local ethnicities, which would extend our model application to new data with ethnicities different from ours.

The prevalence of depressive symptoms in older adults differed among ethnic groups (Table 1). The Tolaki ethnic group had the highest prevalence. Prevalences were similar between the Bugis-Makassar and Buton ethnic groups. Only one local ethnic group was similar to those not from Sulawesi Island in terms of the prevalence estimate of depressive symptoms, which was the Muna ethnic group. Both the Bugis-Makassar and the Tolaki ethnic groups were considered the majority of community-dwelling older adults in our dataset.

We only used the training set to develop the models. This procedure would be similar to prediction model developed and validated under different studies. Nevertheless, we needed to identify the characteristics of the dataset we used for training the prediction models (Table 1). Future use of our models will likely benefit those with similar characteristics, particularly in the predictors used in the final model. As intended, we developed our models for older adults aged ≥60 years. This intention characterizes older adults as reasonably having comorbidities and poorer health conditions of hearing, oral status, and visual function, which are considerable compared to younger adults. However, in all those categorical variables (excluding

**Table 1. Most subjects have not obtained a university education, were not separated/divorced, and are religious believers.**

| Description | | Prevalence (95% CI) | GDS-15 (-) | GDS-15 (+) | *p*-value | Any outcome |
|---|---|---|---|---|---|---|
| **Data partitioning** | | | | | | |
| Training set | | | 393 | 609 | NA | 1002/1252 (80.03%) |
| Test set | | | 97 | 153 | NA | 250/1252 (19.97%) |
| Model development (training and test sets) | Local ethnic groups | | 490 | 762 | NA | 1252/1381 (90.66%) |
| | Tolaki | 64.57% (63.59 to 65.55) | 144 | 260 | 0.012* | 404/1381 (29.25%) |
| | Bugis-Makassar | 61.12% (60.27 to 61.98) | 201 | 321 | 0.048* | 522/1381 (37.8%) |
| | Buton | 59.93% (58.31 to 61.55) | 41 | 58 | 0.318 | 99/1381 (7.17%) |
| | Muna | 54.96% (53.62 to 56.31) | 104 | 123 | 0.683 | 227/1381 (16.44%) |
| Model validation | Non-local ethnic group | 52.2% (50.67 to 53.73) | 62 | 67 | (reference) | 129/1381 (9.34%) |
| Total (development and validation) | | | 552 | 829 | NA | 1381/1381 (100%) |
| **Variables in the training set** | | | | | | |
| Age (years) | | | 66 (± 6) | 66 (± 7) | 0.935 | |
| No. of comorbidities | | | 1 (± 1) | 1 (± 1) | 0.005* | |
| Income (IDR) | | | 1,476,417 (±1,310,653) | 1,354,171 (±1,318,338) | 0.152 | |
| No. of medications | | | 1 (± 1) | 1 (± 1) | 0.198 | |
| No. of children | | | 4 (± 2) | 4 (± 2) | 0.415 | |
| Duration of routine visits to CHCs (years) | | | 9 (± 9) | 9 (± 10) | 0.664 | |
| Educational level | Primary | | 122 (31.04%) | 183 (30.05%) | (reference) | |
| | High school | | 109 (27.74%) | 194 (31.86%) | 0.307 | |
| | Secondary | | 68 (17.3%) | 93 (15.27%) | 0.640 | |
| | Illiterate | | 53 (13.49%) | 64 (10.51%) | 0.323 | |
| | University | | 32 (8.14%) | 54 (8.87%) | 0.640 | |
| | Other | | 9 (2.29%) | 21 (3.45%) | 0.287 | |
| Employed before | Yes | | 230 (58.52%) | 294 (48.28%) | (reference) | |
| | No | | 163 (41.48%) | 315 (51.72%) | 0.002* | |
| Employed now | No | | 319 (81.17%) | 514 (84.4%) | (reference) | |
| | Yes | | 74 (18.83%) | 95 (15.6%) | 0.183 | |
| Gender | Female | | 249 (63.36%) | 345 (56.65%) | (reference) | |
| | Male | | 144 (36.64%) | 264 (43.35%) | 0.035* | |
| Health condition | Fair | | 226 (57.51%) | 349 (57.31%) | (reference) | |
| | Good | | 106 (26.97%) | 204 (33.5%) | 0.134 | |
| | Poor | | 48 (12.21%) | 43 (7.06%) | 0.016* | |
| | Very good | | 9 (2.29%) | 11 (1.81%) | 0.609 | |
| | Very poor | | 4 (1.02%) | 2 (0.33%) | 0.195 | |
| Hearing problems | No | | 303 (77.1%) | 485 (79.64%) | (reference) | |
| | Yes | | 90 (22.9%) | 124 (20.36%) | 0.338 | |
| Living status | Living with spouse and other family members | | 221 (56.23%) | 330 (54.19%) | (reference) | |

(*Continued*)

**Table 1.** (Continued)

| Description | | Prevalence (95% CI) | GDS-15 (-) | GDS-15 (+) | *p*-value | Any outcome |
|---|---|---|---|---|---|---|
| | Living with family members but without a spouse | | 111 (28.24%) | 145 (23.81%) | 0.383 | |
| | Living with a spouse only | | 45 (11.45%) | 82 (13.46%) | 0.331 | |
| | Living alone | | 16 (4.07%) | 51 (8.37%) | 0.011* | |
| | Other | | 0 (0%) | 1 (0.16%) | 0.970 | |
| Marital status | Married | | 273 (69.47%) | 419 (68.8%) | (reference) | |
| | Widowed | | 113 (28.75%) | 168 (27.59%) | 0.826 | |
| | Separated/divorced | | 4 (1.02%) | 19 (3.12%) | 0.042* | |
| | Single | | 3 (0.76%) | 3 (0.49%) | 0.601 | |
| Oral status | Fair | | 218 (55.47%) | 355 (58.29%) | (reference) | |
| | Good | | 129 (32.82%) | 198 (32.51%) | 0.677 | |
| | Poor | | 42 (10.69%) | 48 (7.88%) | 0.121 | |
| | Very good | | 4 (1.02%) | 6 (0.99%) | 0.900 | |
| | Very poor | | 0 (0%) | 2 (0.33%) | 0.972 | |
| Religion | Moslem | | 389 (98.98%) | 598 (98.19%) | (reference) | |
| | Christian | | 4 (1.02%) | 11 (1.81%) | 0.322 | |
| | Hindu | | 0 (0%) | 0 (0%) | NA | |
| Visual problems | Yes | | 200 (50.89%) | 318 (52.22%) | (reference) | |
| | No | | 192 (48.85%) | 291 (47.78%) | 0.712 | |
| | Missing | | 1 (0.25%) | 0 (0%) | NA | |

* *p* value <0.05; 95% confidence interval was estimated by 30-times bootstrapping. GDS-15, 15-Item Geriatric Depression Scale; NA, not applicable; IDR, Indonesian Rupiah; CHC, community health center.

comorbidities), most were in a fair health condition, probably because these subjects had routinely visited a CHC for 9 or 10 years on average. Most of the subjects had not obtained a university education. They had two to six children, were mostly not separated/divorced, most lived with either a spouse or other family members, and were unemployed. Their incomes were considerably low for this country. Most of the subjects, if not all, were religious believers. We saw similar characteristics between GDS-15 positives and negatives, except for the Tolaki (*p* = 0.012) and Bugis-Makasar ethnic groups (*p* = 0.48), the number of comorbidities (*p* = 0.005), the employment status before 60 years of age (*p* = 0.002), male gender (*p* = 0.035), a poor health condition (*p* = 0.016), a living alone status (*p* = 0.011), and a separated/divorced status (*p* = 0.042). In addition, to deploy our models, we provided a web application (https://predme.app/pre_gds15) using the best models as a prototype before incorporating the application into an EHR system. Religion options were provided for many religions to keep the application inclusive and avoid inequality. We also used a Big Mac index commonly used to convert income to the same notion in any country [69].

## The well-calibrated models were SPC-GBM with re-calibration and DI-VNN without re-calibration

We applied binarization of categorical variables of 17 predictors resulting in 37 predictors without a perfect separation problem in the training set. We used these predictors to develop an LR model with regularization. Because we needed to pursue 20 events per variable, only the top 19 PCs were retained for feature selection by the multivariable LR. These PCs accounted for 81.7% of the variance explained (95% CI 81.68% to 81.72%). Furthermore, we only used

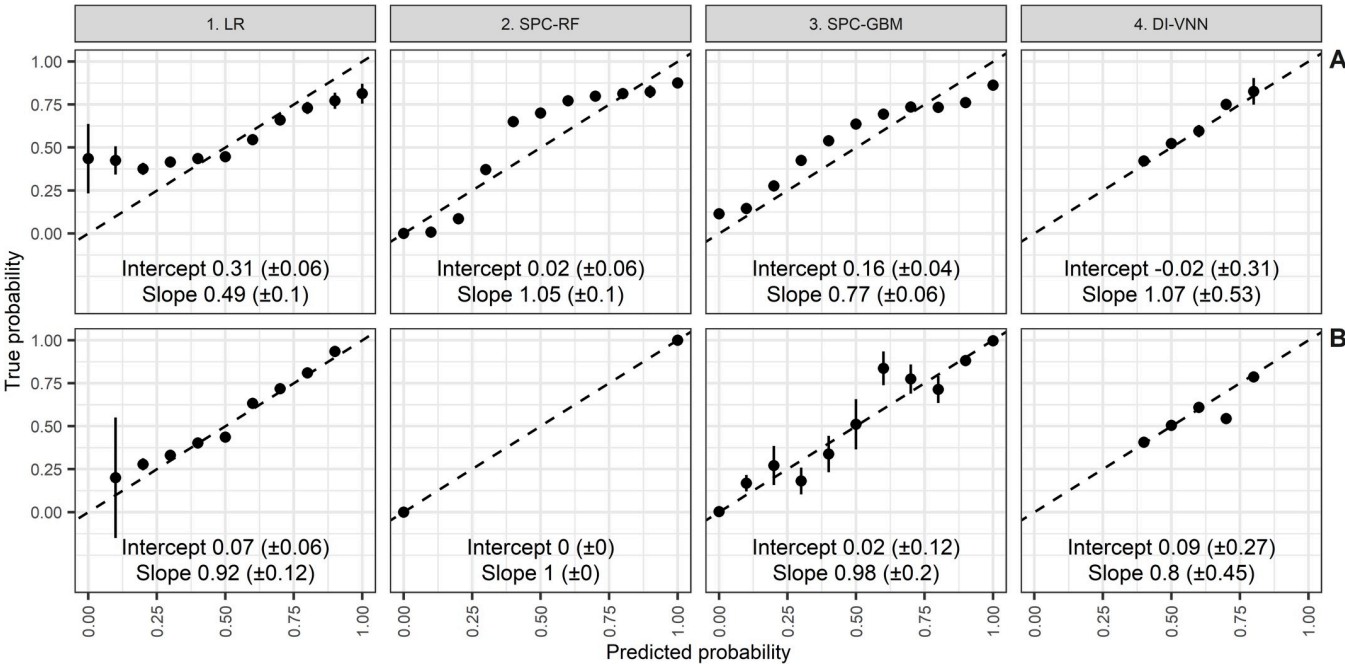

**Fig 2. The well-calibrated models were SPC-GBM with re-calibration and DI-VNN without re-calibration.** A. Without re-calibration. B. With re-calibration. DI-VNN, deep-insight visible neural network; LR, logistic regression; SPC-GBM, selected principal components with the gradient boosting machine; SPC-RF, selected principal components with the random forest.

seven selected PCs for model development by the RF and GBM algorithms because this allowed us to pursue >50 events per variable. These were the selected PC (SPC)-RF and SPC-GBM models. Meanwhile, of 17 predictors and 37 PCs for the DI-VNN, only 18 had an FDR of <0.05 by the differential analysis using the Benjamini-Hochberg correction. This analysis pre-selected all candidate predictors before being a candidate predictor of the DI-VNN. In the differential analysis, only one variable was used for each analysis. This procedure ensured more than 20 events per variable for each analysis. Subsequently, the Benjamini-Hochberg method corrected the multiple testing effects. We compared calibration metrics and plots of these models with and without re-calibration (Fig 2). Only two models were well-calibrated. These were SPC-GBM with re-calibration (Fig 2B) and the DI-VNN without re-calibration (Fig 2A). The LR model was visually aligned after re-calibration (Fig 2B), but the 95% CI of the calibration intercept did not cover 0. The SPC-RF without re-calibration also did not cover 1 by the 95% CI of the calibration slope. Meanwhile, re-calibrating this model resulted in a dichotomous probability that reduced its clinical utility (Fig 2B). Neither the calibration intercept nor slope of the SPC-GBM without re-calibration (Fig 2A) respectively covered 0 or 1. Unlike this model, the DI-VNN with re-calibration (Fig 2B) fulfilled the intercept and slope criteria but not the calibration plot.

## The best model was the SPC-GBM but undifferentiated from the DI-VNN in external validation

We only used the training set to determine the best from two well-calibrated models: SPC-GBM with re-calibration (Table 2). As observed in this study, the RF and GBM algorithms achieved suitable predictive performances by overfitting the training set. For example, predictive performances of SPC-GBM were reduced by 42.08% and 37.98%, respectively, for

**Table 2. The best model was the SPC-GBM but undifferentiated from the DI-VNN in external validation.**

| Data partition | Model | AUROC (95% CI) | Threshold (95% CI)* | Metric | Value (95% CI) |
|---|---|---|---|---|---|
| Internal validation, modeling | SPC-GBM, with re-calibration | 0.998 (0.998 to 0.998) | 0.95 (0.948 to 0.952) | Sensitivity | 0.900 (0.897 to 0.903) |
| | | | | Specificity | 0.997 (0.997 to 0.997) |
| | | | | Accuracy | 0.949 (0.947 to 0.950) |
| | | | | NPV | 0.909 (0.906 to 0.912) |
| | | | | PPV | 0.997 (0.996 to 0.997) |
| | DI-VNN, without re-calibration | 0.605 (0.605 to 0.606) | 0.205 (0.198 to 0.212) | Sensitivity | 0.996 (0.995 to 0.997) |
| | | | | Specificity | 0.008 (0.006 to 0.010) |
| | | | | Accuracy | 0.502 (0.501 to 0.503) |
| | | | | NPV | 0.655 (0.652 to 0.658) |
| | | | | PPV | 0.501 (0.500 to 0.502) |
| External validation, local ethnicity | SPC-GBM, with re-calibration | 0.578 (0.572 to 0.583) | 0.95 (0.948 to 0.952) | Sensitivity | 0.637 (0.631 to 0.643) |
| | | | | Specificity | 0.510 (0.504 to 0.517) |
| | | | | Accuracy | 0.573 (0.569 to 0.578) |
| | | | | NPV | 0.588 (0.582 to 0.594) |
| | | | | PPV | 0.562 (0.556 to 0.569) |
| | DI-VNN, without re-calibration | 0.577 (0.576 to 0.579) | 0.205 (0.198 to 0.212) | Sensitivity | 0.997 (0.996 to 0.998) |
| | | | | Specificity | 0.005 (0.003 to 0.006) |
| | | | | Accuracy | 0.498 (0.497 to 0.500) |
| | | | | NPV | 0.613 (0.606 to 0.620) |
| | | | | PPV | 0.498 (0.496 to 0.499) |
| External validation, non-local ethnicity | SPC-GBM, with re-calibration | 0.619 (0.610 to 0.627) | 0.95 (0.948 to 0.952) | Sensitivity | 0.566 (0.556 to 0.575) |
| | | | | Specificity | 0.607 (0.599 to 0.616) |
| | | | | Accuracy | 0.587 (0.581 to 0.593) |
| | | | | NPV | 0.593 (0.584 to 0.602) |
| | | | | PPV | 0.581 (0.572 to 0.589) |
| | DI-VNN, without re-calibration | 0.579 (0.576 to 0.581) | 0.205 (0.198 to 0.212) | Sensitivity | 0.994 (0.992 to 0.995) |
| | | | | Specificity | 0.011 (0.008 to 0.013) |
| | | | | Accuracy | 0.486 (0.484 to 0.489) |
| | | | | NPV | 0.631 (0.625 to 0.637) |
| | | | | PPV | 0.485 (0.483 to 0.488) |

* Thresholds closer to a sensitivity of 0.90 by the training set were taken from each bootstrapped subset. AUROC, the area under the receiver operating characteristics curve; NPV, negative predictive value; PPV, positive predictive value; SPC-GBM, selected positive components of the gradient-boosting machine; DI-VNN, deep-insight visible neural network.

point estimates of AUROCs using validation sets with local (0.578, 95% CI 0.572 to 0.583) and non-local (0.619, 95% CI 0.610 to 0.627) ethnicities, compared to those using training set (0.998, 95% CI 0.998 to 0.998). However, the models developed using these algorithms often outperformed models using other algorithms in external validation. That was not always the case in this study. Predictive performances of the SPC-GBM (AUROC of 0.578, 95% CI 0.572 to 0.583; $n$ = 250) were similar to those of the DI-VNN without re-calibration (AUROC of 0.577, 95% CI 0.576 to 0.579; $n$ = 250) in an external validation set with a local ethnic group. In addition, the DI-VNN also showed similar predictive performances among those using training (AUROC of 0.577, 95% CI 0.576 to 0.579; $n$ = 1002) and two test sets, either with a local (AUROC of 0.577, 95% CI 0.576 to 0.579; $n$ = 250) or non-local (AUROC of 0.577, 95% CI 0.576 to 0.579; $n$ = 129) ethnic group. These predictive performances were achieved, although we considered the number of events per variable under different principles from those of other

models. In addition, a previous study also applied a questionnaire-free method to predict the GDS-15 in older adults living alone using a wearable device, but the model was considerably overfitting because of a small sample size (AUROC 0.96, 95% CI 0.91 to 0.99; *n* = 47) [24]. In addition, according to any metrics evaluated in this study, predictive performances of SPC-GBM were better than random or coin-flip guessing (e.g., the AUROC point estimate of SPC-GBM >0.5).

## A low education but literate and living alone was predictive in the SPC-GBM while living alone with significant life events, religion, and family support were predictive in the DI-VNN

Using both models, we could identify how important the predictors are in predicting the GDS-15. There were seven PCs in the SPC-GBM. They were latent variables that represented the 37 predictors but with different weights. Details are described elsewhere on how the weights were inferred [62]. We visualized the absolute values of these weights for each selected PC (Fig 3). Absolute values were used because the positive/negative values cannot be interpreted straightforwardly, regardless of whether these tend to be events or non-events. By observing the visualization, we could infer the meaning of the latent variables. These were named based on the higher absolute values by referring to particular predictors.

The most important PC in the SPC-GBM was PC11 (education and living status). In this PC, older adults with a low education but literate and living alone tended to be predictive. The other most important PCs were PC4, PC8, and PC10, which implied religious perceptions, educational perceptions, and current employment status on health. We should have described religion explicitly to maintain our prediction models' inclusiveness. Education also contributes to PC10. Both PC8 and PC10 also had larger weights on the oral status. Less important predictors were PC16 (very poor hearing), PC14 (very poor health and others), and PC18 (unknown). The last PC has sporadic, slightly weighted predictors.

The DI-VNN also selected PC4 and PC18 as features (Fig 3). There were original predictors selected in this model, which were religion A (F1) or Z (F3), poor (F2) or good (F10) health conditions, living alone (F6) or with family members but without a spouse (F8), a separated/divorced marital status (F7), a previously employed status (F9), medications (F4), and comorbidities (F5). Beyond PC4 and PC18, there was PC5 (health problems). It was related to comorbidities (F5) and medications (F4). PC4 was also reinforced by PC37 (religion), with less involvement in the health aspect. Poor-health medication (PC28) was also selected with larger weights on the selected predictors, which were a poor health condition (F2) and medications (F4), and the deselected ones, which were education and income. PC27 and PC26 were the previous employment status (F9), but PC27 also had larger weights on age, hearing problems, and the number of children. The last PC21 had larger weights on several predictors related to family support of health.

## Living alone with significant life events was positively predictive in the DI-VNN but the opposite if believing in a religion that attracts family activities

While PCs in the SPC-GBM were independently interpreted, those could be interconnected in the DI-VNN (Fig 4). It also included the predictors of origin. Each ontology predicted an outcome in the DI-VNN, contributing to the optimization of the predictive performance. If we used the model architecture up to each ontology for predicting the outcome, different AUROCs were shown (Fig 4A). The top three highest AUROCs were those predicted up to the root, ONT:20, and ONT:22. Each ontology was visualized for the array difference between GDS-15

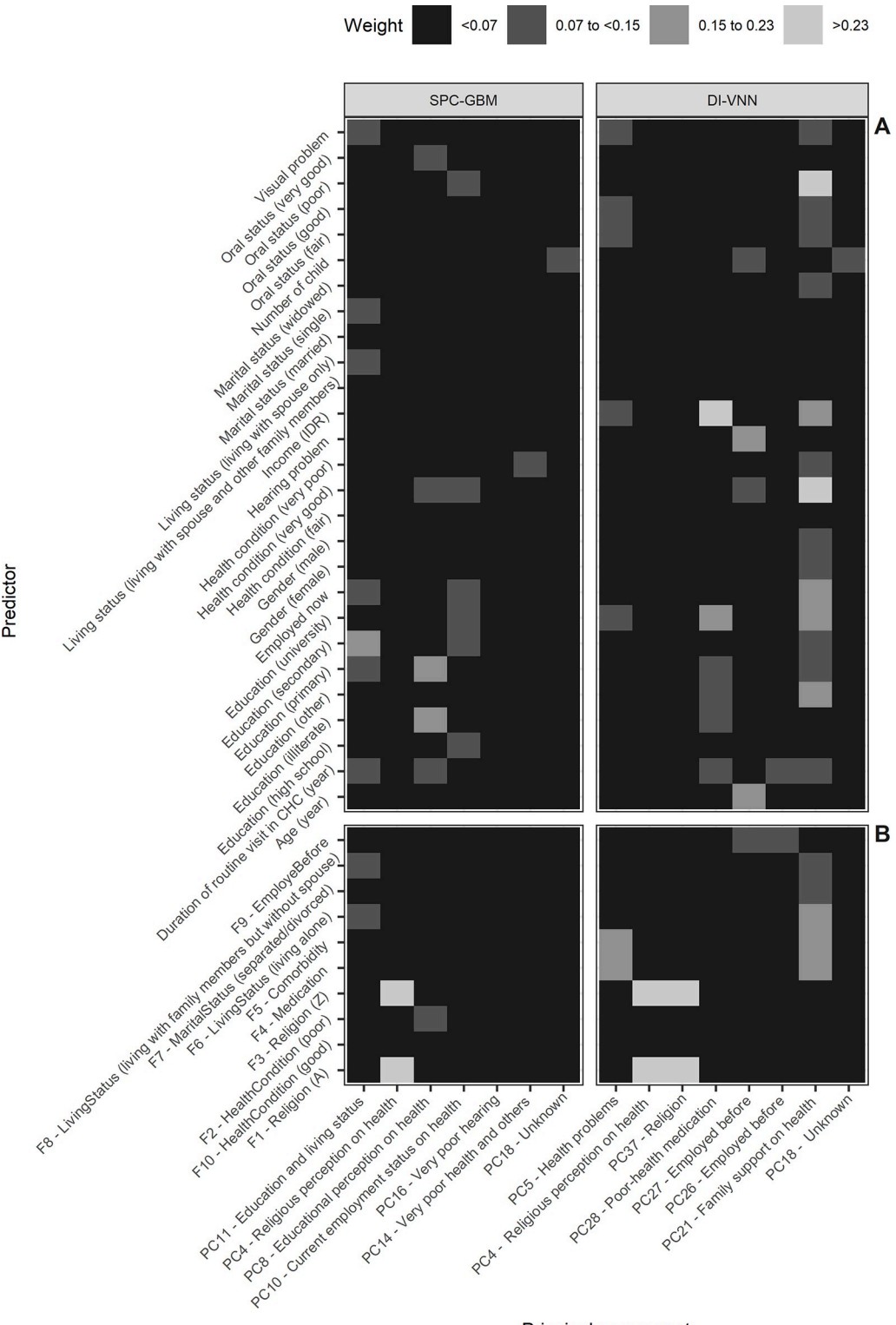

**Fig 3. A low education but literate and living alone was predictive in the SPC-GBM while living alone with significant life events, religion, and family support were predictive in the DI-VNN.** A. Not selected by the DI-VNN. B. Selected by the DI-VNN. CHC, community health center; DI-VNN, deep-insight visible neural network; IDR, Indonesian Rupiah; LR, logistic regression; PC, principal component; SPC-GBM, selected PCs with the gradient boosting machine.

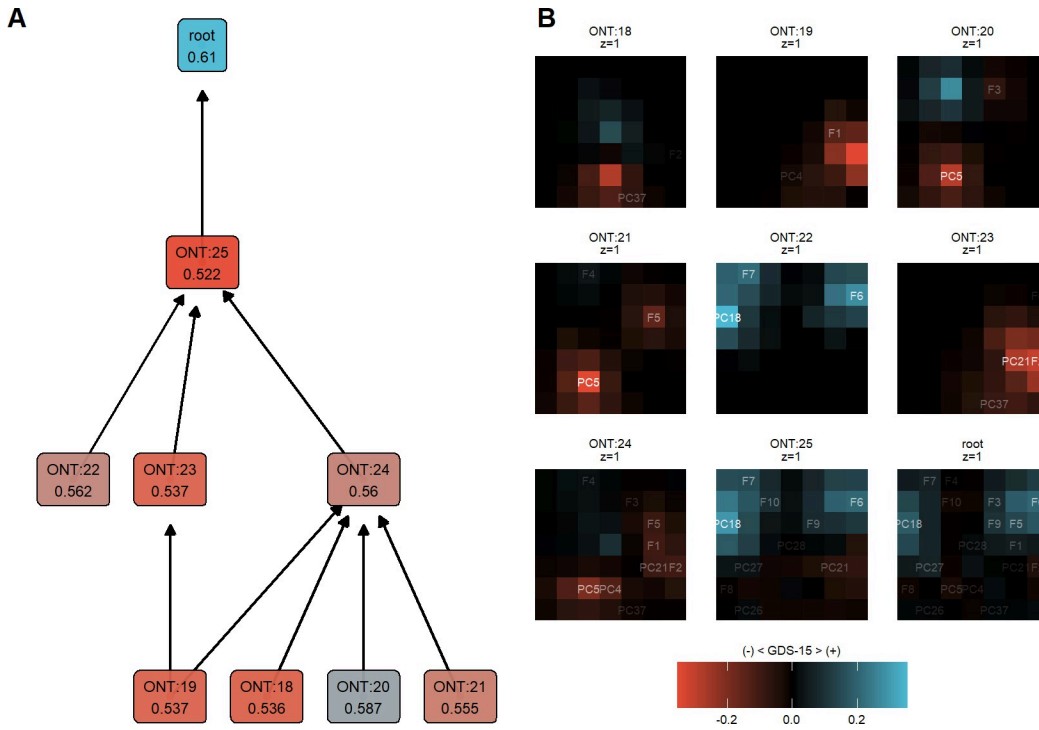

**Fig 4. Living alone with significant life events was positively predictive in the DI-VNN but the opposite if believing in a religion that attracts family activities.** A. Ontology network. B. Ontology array. The number under the ontology name in Fig 4B is the area under the receiver operating characteristics curve (AUROC). Please see Fig 3 for the annotation of predictors in Fig 4B. DI-VNN, deep-insight visible neural network; GDS-15, 15-item geriatric depression scale; ONT, ontology; z, channel.

positives and negatives (Fig 3B). Those for the negatives subtract the weighted features for GDS-15 positives. Positive and negative results from this subtraction referred to GDS-15 positive and negative predictions. Details are described elsewhere on how each ontology prediction was taken into the final prediction and which layers were used for feature visualization [65].

In the root ontology array, subjects living alone (F6) with comorbidities (F5) and multiple factors (PC18, unknown) tended to be predicted as GDS-15 positives. By tracing through ONT:25 and ONT:22, the PC18 factors were closer to the separated/divorce marital status (F7). Health problems (PC5) in ONT:20 were also closer to the religious perception of health (PC4) in the parent ontology, which was ONT:24. Subjects with PC5 and PC4 tended to be predicted as GDS-15 negatives. Similar predictions assigned subjects with family support (PC21) on poor health conditions (F2), as shown by ONT:23. This was related to religion A (F1) in ONT:19 that was connected to ONT:24 with PC21, F2, and F5 (comorbidities). The last feature in ONT:24 had an opposite tendency on the GDS-15 outcome with the same feature in the root ontology.

## Discussion

In this study, we developed four machine-learning models to predict GDS-15 results among community-dwelling older adults. Experimental results demonstrated the feasibility of our approach of applying a questionnaire-free method for developing a triage test for the GDS-15 based on routine data from CHCs. The predictive performances were validated using random and non-random data partitioning, but we only used the training set to develop the models.

The validation allowed model generalization to a non-local ethnic group for the SPC-GBM and DI-VNN models.

From 37 PCs, we found seven PCs with top absolute weights, which contributed to the prediction using the SPC-GBM with re-calibration. In comparison, 10 original predictors and eight PCs contributed to the prediction using the DI-VNN without re-calibration. A web application is provided using both the SPC-GBM and DI-VNN, but the latter model was used for individual exploration of either protective or risk factors. It is because the DI-VNN has a deeper exploration capability. In this study, the AUROC of the DI-VNN was very similar to that of the SPC-GBM. This finding may reveal insight into precise behavioral interventions: (1) to prevent depressive symptoms from turning into major depressive disorders; or (2) to mitigate further progression of this disorder. Note that our system should also be automatic in recommending a GDS-15 evaluation. Manual input by clinicians considerably cancels out the objective of this predictive system.

SPC-GBM has demonstrated moderate sensitivity and specificity based on external validation with either local or non-local ethnicity. Among individuals experiencing depressive symptoms (i.e., positives), an incorrect prediction (i.e., a negative) may cause an individual to be undiagnosed. Hence, a predicted negative should be confirmed by DI-VNN, which demonstrated high sensitivity by external validation. Contrarily, among individuals without depressive symptoms, this may cause overdiagnosis. However, the false positives will be screened by GDS-15 instead of being a definitive diagnosis. Nonetheless, using the baseline value, the predictive performance of SPC-GBM was better than random or coin-flip guessing for any metrics evaluated in this study.

Several findings in our study were in line with previous studies. These included education [33, 34], living status [38–40], religion [35, 36], previous employment [29–31], health status [41–43], and current employment [31, 32]. In PC11, older adults with low education but literate and living alone were strong predictors of GDS-15 positives. However, our findings contradict previous findings that reported education was negatively correlated with depressive symptoms (Xin and Ren, 2020) [70].

The second predictor was religious perceptions of health in PC4, with important predictors having a fair oral health status, a fair health condition, and religious beliefs. Depressive symptoms were associated with religiosity based on these PCs. This finding is in line with those of previous studies, which showed that the severity of depression increased with a higher number of missing teeth, the number of decayed teeth, and oral dryness [45]. In addition, religious beliefs were among the important variables in our prediction models. Faithful religious believers have lower levels of depression than non-believers [36].

The educational perception of health (PC8) was also an important predictor of depressive symptoms, which consisted of a very good oral health status, having a primary education and being illiterate, having poor/very good health, and the duration of routine visits to the CHC. The frequency of regular visits to the CHC in this study might have promoted good health in these older adults. CHCs are the first-line promoter of community health, and this seems to be protective against depression. However, depression was also associated with the length of stay, outpatient and inpatient costs, and increasing use of any healthcare facility, including outpatient visits [43].

Another important variable was PC10, which consisted of poor oral health status, very good health conditions, current employment, and higher education. Unemployed individuals and individuals who moved from permanent to precarious employment had an increased risk of clinically relevant depression [32, 71]. Nevertheless, among older people who work, depression can also cause job loss [30]. Therefore, the predictive value of this latent variable may be either a cause or an effect of depressive symptoms.

Very poor hearing in PC16 was also important for predicting depressive symptoms. It is reasonable that hearing problems and very poor health conditions would increase the risk of depressive symptoms. Hearing loss is the third most frequent chronic health problem among older adults and can affect health conditions [72, 73]. The low health conditions of PC14 were also the same as those of PC5 with health problems of comorbidities and medications (PC28). Lastly, we found other important variables in the DI-VNN model: 1) family support of health (PC21) with predictors of the oral status and visual problems, health conditions and income, education, employment, and gender; 2) living (F6, F8) and marital status (F7); and 3) comorbidities (F5) and medications (F4). Income was also a determinant factor of depression in outpatient care in hospitals in Indonesia [34].

In conclusion, the best prediction models were the SPC-GBM and DI-VNN models. One can use these models in our web application to screen for depressive symptoms along with the GDS-15 at any time. If deemed positive, according our models, an older adult is only then asked to answer questions in the GDS-15. This workflow allows for more-frequent screening and may help detect depressive symptoms at any time. Since later-life depression often causes multiple physical symptoms, we would expect reduced unnecessary costs for related diagnostic procedures and interventions. However, future studies are needed to confirm the impacts of our models in improving both the detection and early intervention of older adults with depression.

## Limitations of the study

This study has several limitations. An older adult who is an atheist or believes in religion beyond those in our dataset might not be well-predicted. The Big Mac index perceives income as a notion of primary need, which is food, while depressing problems related to income may manifest as different notions. Populations with similar characteristics to those in our training set are warranted to use our prediction models. The predictive performance may differ if older adults have high education, are single, have previous employment, have a job, and have no religious beliefs. More-similar characteristics to our target population would lead to more-optimal predictive performance.

Although the SPC-GBM with re-calibration had the best performance in the internal validation set among the well-calibrated models, the performances were undifferentiated in the external validation set with local ethnicity compared to the DI-VNN without re-calibration. Nonetheless, we only used the internal validation set to choose the best model. It is because choosing the best model by the external validation set might lead to an optimistic bias or overfitting; instead, external validation sets were used for a robustness test of the performance of the prediction models [58]. Eventually, despite the model's reliability demonstrated in the paper using external validation, one should still not assume generalizability for any other population with different characteristics. External validation is still required for such population. Yet, this is a general issue in prediction studies, not limited to our study.

## Inclusion and diversity

We worked to ensure gender balance in the recruitment of human subjects. We worked to ensure ethnic or other types of diversity in the recruitment of human subjects. We worked to ensure that the study questionnaires were prepared in an inclusive way. One or more of the authors of this paper self-identifies as an underrepresented ethnic minority in science. The author list of this paper includes contributors from the location where the research was conducted who participated in the data collection, design, analysis, and/or interpretation of the work.

## Supporting information

**S1 File. Checklists and questionnaire.** This file consists of: (1) S1 Table. Guidelines for developing and reporting machine learning predictive models in biomedical research; (2) S2 Table. Prediction model risk of bias assessment tools (PROBAST); (3) S3 Table. Clinical checklists for assessing suitability of machine learning applications in healthcare; and (4) S4 Table. The 15-item Geriatric Depression Scale (GDS-15) questionnaire.
(DOCX)

## Author Contributions

**Conceptualization:** Sri Susanty, Herdiantri Sufriyana, Emily Chia-Yu Su, Yeu-Hui Chuang.

**Data curation:** Sri Susanty.

**Formal analysis:** Herdiantri Sufriyana.

**Funding acquisition:** Herdiantri Sufriyana, Emily Chia-Yu Su.

**Investigation:** Sri Susanty.

**Methodology:** Sri Susanty, Herdiantri Sufriyana, Emily Chia-Yu Su, Yeu-Hui Chuang.

**Project administration:** Sri Susanty, Yeu-Hui Chuang.

**Resources:** Sri Susanty, Emily Chia-Yu Su, Yeu-Hui Chuang.

**Software:** Herdiantri Sufriyana.

**Supervision:** Emily Chia-Yu Su, Yeu-Hui Chuang.

**Validation:** Sri Susanty.

**Visualization:** Herdiantri Sufriyana.

**Writing – original draft:** Sri Susanty, Herdiantri Sufriyana.

**Writing – review & editing:** Emily Chia-Yu Su, Yeu-Hui Chuang.

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
