## [Decision Letter · Decision Letter 0]

15 Aug 2022

PONE-D-22-10670Questionnaire-free machine-learning method to predict depressive symptoms among community-dwelling older adultsPLOS ONE

Dear Dr. Chuang,

Thank you for submitting your manuscript to PLOS ONE. After careful consideration, we feel that it has merit but does not fully meet PLOS ONE’s publication criteria as it currently stands. Therefore, we invite you to submit a revised version of the manuscript that addresses the points raised during the review process.

The reviewers' have raised a number of concerns that you should respond to as part of your revision. Please provide further detail as to the added value of the prediction model over currently available clinical tools. The comments from Reviewer 2 are strongly worded, however, it would be beneficial to provide the rationale for the methods you have chosen to employ here and to present your analyses in a fashion that makes these clear and easy to follow. An additional minor comment is that your 'Summary' should be renamed 'Abstract'. Please note that PLOS ONE has specific guidelines on code sharing for submissions in which author-generated code underpins the findings in the manuscript. In these cases, all author-generated code must be made available without restrictions upon publication of the work. Please review our guidelines at https://journals.plos.org/plosone/s/materials-and-software-sharing#loc-sharing-code and ensure that your code is shared in a way that follows best practice and facilitates reproducibility and reuse.

We look forward to receiving your revised manuscript.

Kind regards,

Callam Davidson

Editorial Office

PLOS ONE

Journal Requirements: 

2. Please note that PLOS ONE has specific guidelines on code sharing for submissions in which author-generated code underpins the findings in the manuscript. In these cases, all author-generated code must be made available without restrictions upon publication of the work. Please review our guidelines at https://journals.plos.org/plosone/s/materials-and-software-sharing#loc-sharing-code and ensure that your code is shared in a way that follows best practice and facilitates reproducibility and reuse

3. We note that you have stated that you will provide repository information for your data at acceptance. Should your manuscript be accepted for publication, we will hold it until you provide the relevant accession numbers or DOIs necessary to access your data. If you wish to make changes to your Data Availability statement, please describe these changes in your cover letter and we will update your Data Availability statement to reflect the information you provide

"This work was supported by the Ministry of Science and Technology (MOST) in Taiwan (grant no. MOST109-2221-E-038-018 and MOST110-2628-E-038-001) and the Higher Education Sprout Project from the Ministry of Education (MOE) in Taiwan (grant no. DP2-110-21121-01-A-13) to Emily Chia-Yu Su. The sponsor had no role in the research design or contents of the manuscript for publication."

Reviewers' comments:

Reviewer's Responses to Questions

**Comments to the Author**

1. Is the manuscript technically sound, and do the data support the conclusions?

Reviewer #1: Partly

Reviewer #2: No

2. Has the statistical analysis been performed appropriately and rigorously? 

Reviewer #1: Yes

Reviewer #2: No

3. Have the authors made all data underlying the findings in their manuscript fully available?

Reviewer #1: No

Reviewer #2: Yes

4. Is the manuscript presented in an intelligible fashion and written in standard English?

Reviewer #1: Yes

Reviewer #2: No

5. Review Comments to the Author

Reviewer #1: Positionality: I am a researcher in information science studying the reliability and appropriate use of digital mental health biomarkers. Most of my work has focused on developing and analyzing machine learning models of major depressive disorder, generalized anxiety disorder, and schizophrenia. I have not studied the mental health of older adults specifically.

Overall comment: The authors present a paper to predict depression symptoms of older adults. They develop a variety of machine learning models to predict a self-reported depression questionnaire, and then test these models in two holdout test sets, one from a similar population as the training dataset, and one from a different population. The authors then provide two analyses to try and explain the relationship between the model predictors and the outcome variables.

Overall, I appreciate the authors’ thorough documentation of results for the main prediction analysis, and the focus on older individuals, a population who is deserving of more attention. I do have questions about the motivation, methods and implications that I feel would clarify the validity of the analysis and improve the manuscript. Please see the details below.

Major comments:

1. In the methods it appears that “depressive symptoms” were used as a predictor variable. How was this variable used? Is this current depressive symptoms, or a history of depressive symptoms? If it contains “current depressive symptoms”, then the prediction problem appears trivial, as you’re using a variable in your prediction that represents the outcome you wish to predict. Please clarify and justify its usage.

2. How did the preprocessing methods (e.g. normalization, PCA, multiple imputation chain equation” [MICE]) interact with the 10-fold cross-validation, and testing procedures? Data leakage is a common issue in ML, where data from the holdout data is used within preprocessing. See:

Sayash Kapoor and Arvind Narayanan. 2022. Leakage and the Reproducibility Crisis in ML-based Science. Retrieved July 15, 2022 from http://arxiv.org/abs/2207.07048.

Please confirm data leakage did not occur. If preprocessing models (e.g. mean/standard deviation, MICE model) were created on the entire dataset, I would recommend redeveloping these preprocessing models on each training dataset, and applying them to each held out fold in the revision.

3. How did the authors choose the specific ethnic group to use as the “non-local” external validation set? This specific external validation dataset appeared to have a much more even distribution between GDS-15 positive/negative compared to the training set and test set. Please further justify why the non-local test set is then a good cohort to validate generalization (as stated in the Discussion), and how the differences in outcome distribution between local/non-local data affect the interpretation of the results.

4. Many of the variables used as predictors, and identified as important in the prediction models (e.g. education status, loneliness, deteriorating physical health) are already well-researched variables that are known indicators of mental health symptoms. For example, see:

Susan A Everson, Siobhan C Maty, John W Lynch, and George A Kaplan. 2002. Epidemiologic evidence for the relation between socioeconomic status and depression, obesity, and diabetes. Journal of Psychosomatic Research 53, 4: 891–895.

Evren Erzen and Özkan Çikrikci. 2018. The effect of loneliness on depression: A meta-analysis. International Journal of Social Psychiatry 64, 5: 427–435.

Given many of the variables identified are already known risk factors for depression, what value does the prediction model in the paper add? Please discuss.

5. The authors propose using their model as a screening tool for administering the GDS-15. In the external validation, results showed either moderate sensitivity/specificity (SPC-GBM), or high sensitivity and low specificity (DI-VNN). The authors should add a discussion of how these results impact the usage of each model where either (1) the model will incorrectly classify individuals experiencing depressive symptoms, or (2) there will be a large amount of over prediction.

6. I found the ontology section was unclear; I am not as familiar with ontology-based methods. Could the authors add a section to their methods describing the ontology analysis, and maybe make the relationship between the ontologies, features/PCs, and their underlying meaning more clear in Figure 3?

7. The authors included a link to an online system where clinicians can upload information and the model outputs a prediction, I am assuming using the algorithm in the paper. I am a bit worried about the public nature of this tool, given that the reliability and validity of the tool has not been published, and users could take the prediction model results at face value. I encourage the authors to add a disclaimer about the reliability/validity on the online tool at a basic reading level, so users do not take the prediction result at face value and use it for making care decisions.

In addition, a publication was cited online, which I am assuming is about the prediction model. The links to the publication were not working, and I could not find the publication online. See citation below:

Anonymous, et al. Questionnaire-free method to predict 15-item geriatric depression scale (GDS-15) among community-dwelling elders by machine learning. EBioMedicine (2021). DOI: 00.0000/x00000-000-0000-0 PMID: 00000000 Full text PDF

Could the authors clarify if this is an existing publication, and if so, include it as a supplementary file so we can confirm that the reported results in this manuscript are different from this prior publication?

Finally, when adding my information within the online tool and looking at the GDS-15, I wondered if collecting the information used in the prediction models would really be less burdensome than taking the GDS-15 itself, which is a more direct measure of depression symptoms. In addition, I feared that the tool simply shifts the burden on reporting and entry from the patient to clinicians, who would then need to collect this information for multiple patients, and run the tool. Given this, please justify why a prediction model using these types of demographic data still has utility.

Minor comments:

8. The Authors state that the 15-item GDS is the “most appropriate” version of the scale. What defines “appropriate” in this context? Please clarify in the text.

9. I found the statement “questions asked later in the long term were shown to lead to greater misclassifications” unclear. What do the authors mean? Are they stating that there is a delay in patients with suspected depressive symptoms receiving the questionnaire? Misclassifying what specifically?

10. I found the paragraph of the introduction beginning with “Depression affects 264 million people globally” to cover broader material than the previous paragraph beginning with “Depressive symptoms in older adults”. It might make sense to potentially rearrange these paragraphs to begin with the global burden of depression, then highlight issues with depression questionnaires in older adults. In addition, I believe the summary statistics of depression rates in each country do not add much value to the manuscript. Perhaps the authors could shorten this sentence, or focus on statistics relevant to the specific population studied in the manuscript.

11. The authors state in the Introduction that logistic regression is an insufficient model to develop a triage test for GDS-15 screening, but do not provide reasoning why it is insufficient. An LR model to predict GDS-15 - assuming high sensitivity, specificity, and positive predictive value - would be an ideal model to use due to its explainability, simplicity, and robustness. Please justify further why simple models are insufficient for the specific GDS-15 triage test problem.

12. In the revision, per PLOS ONE’s recommendations, please include the Methods section following the Introduction, before the Results section. Thank you. See: https://journals.plos.org/plosone/s/submission-guidelines

13. How/when was the GDS-15 delivered? During the same screening where the predictor data was collected?

14. In the methods, the authors state: “Meanwhile, over-diagnosis causes an increasing frequency of the use of the GDS-15, which may lead to further misclassification.” Is this true? What does “misclassification” mean in this sense? I believe that administering the GDS-15 after the prediction model would simply validate or mis-validate the prediction model results, not lead to further “misclassification” as the GDS-15 is the “gold standard” in the paper. Maybe the authors could elaborate on other issues that may arise by over-predicting patients experiencing depression symptoms.

15. In the Methods, the authors claim when referring to RF and GBM algorithms: “Both algorithms are the most used competition-winning algorithms for predictions using tabular data.” Do the authors have a citation to back up this claim, and subsequently, why do competition-winning algorithms apply to research and this specific prediction problem? Please provide a better justification.

16. What do the authors mean when they state: “for which characteristics do not imply the data but predict the outcome very well”? Please rewrite this statement for clarity.

17. Why will the data only be available for one year after publication? Can the data be accessed now?

18. The title for the first subsection of the Results states “Most had not obtained a university education, are not separated/divorced, and are religious believers”. Could the authors be more specific on who “Most” refers to?

19. In the results, the authors state “Meanwhile, of 17 predictors and 37 PCs for the DI-VNN, only 18 of them had an FDR of <0.05 by the differential analysis with the Benjamini-Hochberg correction.” What differential analysis did the authors perform? What were the null and alternative hypotheses? Please state in the main text.

20. I would appreciate if the authors stated the AUROC of the best performing models for the external validation in the Results section of the main text. I realize it is on Table 2.

21. What methods were used to identify the important features from the SPC-GBM and DI-VNN models? I know that it is often difficult to extract important features in deep learning algorithms. Thus, I would be interested in how the authors identified the important features in the algorithm.

22. Why do the authors believe there was such significant overfitting in the SPC-GBM model from the internal validation to the external validation cohorts?

Reviewer #2: Detailed Review:

In this paper, the authors present an evaluation of utilizing demographic information as a method for screening for Geriatric Depression. The authors offer a plethora of machine learning models, some more common than others, and evaluate the resulting models to identify which markers were strong indicators of depression in patients. This method would allow care-givers with access to demographic and general factors the opportunity to assess the need to evaluate the patient for depression symptoms.

Key Strength of the paper:

The work is important, instruments that could potentially be used to screen for these kinds of conditions without the immediate input of a patient is highly relevant and an important facet of medical technology

Main Weakness of the paper:

The methodology of this study is very poorly done, or poorly presented. The authors give numbers associated with the models’ results, but the metrics they choose to present are not very meaningful given the context, and further more the authors spend almost no time explaining exactly what types of inputs or how hyper-parameters or even how data-splitting occurred.

The paper is not written very strongly, and there are many details from the implementation and data preparation that are missing. Why did the authors not utilize a cross-fold validation approach? What does internally/externally validated data mean in the context of splitting the data for training and testing? Neural Networks typically require orders of magnitude more data to justify over conventional models such as Decision Trees or Support Vector Machines, why not use those? All of these models are very sensitive to the hyper-parameters you choose, and the types of data you pass in, which were chosen, what was the justification behind it? While the results are generally reported, no meaningful discussion about baselines are provided, how much better than random guessing are these models performing? Why is AUROC the chosen metric, and not F1-score or precision and recall?

These questions are just a few of the many that are left to the reader to try to discern or figure out potentially by having access to the data. While this does not mean that the results presented are not valid ones, there is no understanding of how good they really are? (Would a loaded coin-flip perform better?)

Novelty/Originality, taking into account the relevance of the work for the PLOS ONE audience:

While the domain might be novel, the general approach of using machine learning models in this way is not particularly novel.

Technical/Theoretical Correctness, taking into account datasets, baselines, experimental design, affective theory, are there enough details provided to be able to reproduce the experiments and understand the contribution?

There are a plethora of missing details that are unjustified and under-reported making the correctness of this work difficult to evaluate and exact reproducibility difficult.

Quality of References, is it a good mix of older and newer papers? Do the authors show a good grasp of the current state of the literature? Do they also cite other papers apart from their own work?

The chosen references seem to be reasonable but they way are utilized throughout the paper is not very good. Many things the authors claim go unjustified or uncited, or cited awkwardly. E.g.:

“Nevertheless, the screening frequency of a questionnaire should be limited, because questions asked later in the long term were shown to lead to greater misclassifications (Egleston et al., 2011)”

This sentence doesn't add anything and is unclear. What are "questions asked later in the long term", and how exactly do they lead to greater misclassification? What is the screening frequency that is most appropriate? Does it vary by the depressive state of the individual?

“A triage test with questionnaire-free variables is needed to reduce the frequency of questionnaire use. Demographic and physical health data from routine visits can be utilized.”

This is uncited. Also, what does it mean that it is "needed", wouldn’t a redesigned questionnaire, with repeated evaluation in mind, perform better than one that wasn’t meant to (GDS-15). What about the huge population of people who don’t have access to regular medical attention and physical evaluation, or who might not have access to a good health record/history due to the unavailability of infrastructure for this purpose? Most importantly, none of this is later addressed by references or the methods presented in this work.

While it might be corrected by instead offering the perspective of: “It would allow physicians a more readily and less invasive approach to screen patients for depression risk”

Clarity of Presentation, the English does not need to be flawless, but the text should be understandable.

The presentation of this work is incredibly sub-par. Not only are there a tremendous number of sentences that don’t make sense for logical or grammatical reasons, the discussion of the work at hand is insufficient. It is difficult to follow and the paper repeats itself without clarifying or adding to previous made assertions and conclusions. Overall this is not a well written paper.

6. PLOS authors have the option to publish the peer review history of their article (what does this mean?). If published, this will include your full peer review and any attached files.

Reviewer #1: No

Reviewer #2: No

---

## [Author Response · Author response to Decision Letter 0]

24 Sep 2022

################################################################################

######## 1 ######## Editor

################################################################################

###### 1.1 ###### The reviewers' have raised a number of concerns that you should respond to as part of your revision. Please provide further detail as to the added value of the prediction model over currently available clinical tools. The comments from Reviewer 2 are strongly worded, however, it would be beneficial to provide the rationale for the methods you have chosen to employ here and to present your analyses in a fashion that makes these clear and easy to follow.

Response:

Thank you for giving us a chance to revise this manuscript. Since this study is not a typical diagnostic prediction study, as appreciated by reviewer 2 (see comments 3.1.1 and 3.3.2), it is easily misunderstood to be generalized as the typical ones. Nonetheless, we admit that our wording was insufficient for the core concept of this study, leading to a common misunderstanding for both reviewers. We have provided further detail for the added value in our response to reviewer 1 (see comment 2.2.2) and reviewer 2 (see comment 3.5.2.2).

Concerns from reviewer 2 are welcome and understandable, as our responses to comments 3.2.1 and 3.3.1. Previously, we targeted clinical audience for this manuscript considering its novelty in the domain knowledge. Unfortunately, because of multidisciplinary nature of this study, which was also the reason of why PLOS One, writing the manuscript is challenging in order to be readable for both clinical and computational audiences. Field-specific completeness for one may be considerably distracting for another. We also understand the challenges for finding the peer reviewers with both clinical and computational background, which we appreciate for the editor in this journal.

###### 1.2 ###### An additional minor comment is that your 'Summary' should be renamed 'Abstract'.

Response:

Thank you for reminding us. We have renamed ‘Summary’ to ‘Abstract’.

###### 1.3 ###### Please note that PLOS ONE has specific guidelines on code sharing for submissions in which author-generated code underpins the findings in the manuscript. In these cases, all author-generated code must be made available without restrictions upon publication of the work. Please review our guidelines at https://journals.plos.org/plosone/s/materials-and-software-sharing#loc-sharing-code and ensure that your code is shared in a way that follows best practice and facilitates reproducibility and reuse. We note that you have stated that you will provide repository information for your data at acceptance. Should your manuscript be accepted for publication, we will hold it until you provide the relevant accession numbers or DOIs necessary to access your data. If you wish to make changes to your Data Availability statement, please describe these changes in your cover letter and we will update your Data Availability statement to reflect the information you provide

Response:

Thank you for pointing the potential misunderstanding on code availability. Previously, no restriction was made for the code, although we delayed the data availability. This is because we also had used the data as a part of another study under review elsewhere. Therefore, we have changed from “Data and code availability” to “Data availability statement” with the updated statement, as shown below:

All relevant data files are publicly available from the Github database (https://github.com/herdiantrisufriyana/colab_dep).

###### 1.4 ###### We note that you have provided funding information that is not currently declared in your Funding Statement. However, funding information should not appear in the Acknowledgments section or other areas of your manuscript. We will only publish funding information present in the Funding Statement section of the online submission form. Please remove any funding-related text from the manuscript and let us know how you would like to update your Funding Statement. Please include your amended statements within your cover letter; we will change the online submission form on your behalf.

Response:

We apologize for putting the funding information under incorrect section. We have removed any funding-related text from the manuscript. Since there is no acknowledgement, this section has also been removed. The funding information is amended and has been included within the cover letter. Please kindly change the Funding Statement section of the online submission from.

 

################################################################################

######## 2 ######## Reviewer 1

################################################################################

###### 2.1 ###### Major comments

#### 2.1.1 #### In the methods it appears that “depressive symptoms” were used as a predictor variable. How was this variable used? Is this current depressive symptoms, or a history of depressive symptoms? If it contains “current depressive symptoms”, then the prediction problem appears trivial, as you’re using a variable in your prediction that represents the outcome you wish to predict. Please clarify and justify its usage.

Response:

Thank you for pointing this issue. We did not used “depressive symptoms” as a predictor variable, as currently stated: “… 17) medication (number of prescribed drugs); 18) ethnicity (Bugis-Makassar/Buton/Muna/Tolaki/non-local ethnicity); and 19) depressive symptoms (no/yes). The first 17 attributes were used as predictors.” But, we realized this is misunderstood due to wording. Thus, we revised the paragraph, as shown below:

The dataset consisted of 19 attributes which were 17 candidate-predictor variables, 1 grouping variable, and 1 outcome variable. The candidate-predictor variables were: 1) age (years); 2) gender (male/female); 3) religious beliefs (Christian/Hindu/Moslem); 4) educational attainment (illiterate/primary/secondary/high school/university/other); 5) marital status (single/married/separated or divorced/widowed); 6) children (number of persons); 7) living status (alone/with a family member but no spouse/with a spouse only/with family member and spouse/other); 8) currently employed (no/yes); 9) previously employed (no/yes); 10) income (in Indonesian rupiah (IDR)); 11) duration of visiting the CHC (in the number of years of routine visits); 12) comorbidities (number of conditions); 13) health condition (very good/good/fair/poor/very poor); 14) hearing problems (no/yes); 15) visual problems (no/yes); 16) oral status (very good/good/fair/poor/very poor); and 17) medication (number of prescribed drugs). We used ethnicity (Bugis-Makassar/Buton/Muna/Tolaki/non-local ethnicity) as a grouping variable for data partitioning in order to develop and validate our predictive models (see "Model Validation"). The outcome variable was depressive symptoms (no/yes), as defined in the next section.

#### 2.1.2 #### How did the preprocessing methods (e.g. normalization, PCA, multiple imputation chain equation” [MICE]) interact with the 10-fold cross-validation, and testing procedures? Data leakage is a common issue in ML, where data from the holdout data is used within preprocessing. See: Sayash Kapoor and Arvind Narayanan. 2022. Leakage and the Reproducibility Crisis in ML-based Science. Retrieved July 15, 2022 from http://arxiv.org/abs/2207.07048. Please confirm data leakage did not occur. If preprocessing models (e.g. mean/standard deviation, MICE model) were created on the entire dataset, I would recommend redeveloping these preprocessing models on each training dataset, and applying them to each held out fold in the revision.

Response:

We really appreciate this comment. It points out our effort to rigorously handle outcome leakage in this study in order to follow the guidelines for developing and reporting machine learning predictive models in biomedical research (Luo et al., 2016) (see Table S1). Furthermore, this was also applied at the level of data collection. To avoid outcome leakage, we limited several procedures using only training set (i.e. data partitioning for model development), in addition to careful definition of outcome, which did not include predictor information, vice versa. The outcome-leakage handling was directly stated after each procedure explanation across sections. Yet, we agree with reviewer that such approach of reporting will be easily missed by future readers, and the wording should be improved to keep future readers aware with the outcome-leakage handling effort. Here below we aggregated all the statements (with this comment annotating them in the revised manuscript) and these were also included into Table S5:

A participant answered the questions with assistance of a trained assessor. The GDS questionnaire is described in Table S4. The trained assessor that assisted a participant to fill out the GDS questionnaire was also blinded to the predictor information. Predictor data were demographic data and routine physical health check results from other healthcare givers without knowing the assessment results of depressive symptoms.

We only used the mean and SD calculated from data partitioning for model development. Numerical predictors in any data partitions were standardized using these values.

We imputed the missing value using multiple imputation by the chain equation method after data transformation using only data in the same data partition. Randomly, the missing value was a part of data partitioning for model development.

We only used data partitioning for model development to conduct predictor extraction, representation, and selection.

A ten-fold cross-validation procedure was applied on only data partitioning for model development. We used average values computed from ten rotated matrices of PCs to represent 37 binarized and numerical predictors into 19 PCs. We also used average values of data partitioning for model development to get those PCs for model validation. Details on a resampled dimensional reduction method for this study was already described elsewhere (Sufriyana et al., 2021).

References:

##### Luo, W., Phung, D., Tran, T., Gupta, S., Rana, S., Karmakar, C., Shilton, A., Yearwood, J., Dimitrova, N., Ho, T.B., et al. (2016). Guidelines for Developing and Reporting Machine Learning Predictive Models in Biomedical Research: A Multidisciplinary View. J Med Internet Res 18, e323. 10.2196/jmir.5870.

##### Sufriyana, H., Wu, Y.W., and Su, E.C.Y. (2021b). Resampled dimensional reduction for feature representation in machine learning. Protocol Exchange rs.3.pex-1636/v1. 10.21203/rs.3.pex-1636/v1.

#### 2.1.3 #### How did the authors choose the specific ethnic group to use as the “non-local” external validation set? This specific external validation dataset appeared to have a much more even distribution between GDS-15 positive/negative compared to the training set and test set. Please further justify why the non-local test set is then a good cohort to validate generalization (as stated in the Discussion), and how the differences in outcome distribution between local/non-local data affect the interpretation of the results.

Response:

Thank you for this suggestion. Reviewer has pointed out a potential distrust of future readers to our choice of set for external validation (e.g. cherry picking). Our choice was not driven by the distribution of GDS-15 positive/negative, of which more-even distribution is considerably easier to achieve higher predictive performance. Yet, we admit more explanation is needed to justify the choice of set for external validation and avoid such potential distrust, and to explain how the distribution difference affects the result interpretation, as shown below: 

Model validation may be challenging since estimates of depressive symptom prevalences in the validation set differed from those of the development sets. However, to some extent, ethnicity may affect the distributions of predictors and outcome. A prediction model should be robust against the shift of data distribution, or well-generalized. Therefore, our validation sets allowed the generalization test, including data with non-local ethnicities which would extend our model application to new data with ethnicities different to those in our data.

The model was expected to be used in settings not limited to those with only the local ethnicities. Hence, we should test whether the model developed by using data with local ethnicities would also have an acceptable predictive performance if the model was applied on data with non-local ethnicities.

#### 2.1.4 #### Many of the variables used as predictors, and identified as important in the prediction models (e.g. education status, loneliness, deteriorating physical health) are already well-researched variables that are known indicators of mental health symptoms. For example, see: (1) Susan A Everson, Siobhan C Maty, John W Lynch, and George A Kaplan. 2002. Epidemiologic evidence for the relation between socioeconomic status and depression, obesity, and diabetes. Journal of Psychosomatic Research 53, 4: 891–895; and (2) Evren Erzen and Özkan Çikrikci. 2018. The effect of loneliness on depression: A meta-analysis. International Journal of Social Psychiatry 64, 5: 427–435. Given many of the variables identified are already known risk factors for depression, what value does the prediction model in the paper add? Please discuss.

Response:

Thank you for identifying potential misconception to this study by future readers. We have addressed this comment, as shown below:

Notably, we conducted this study for the objective of predicting depressive symptoms, instead of finding a risk factor. While risk factors in our model were already identified by the previous studies, each risk factor is only able to estimate likelihood of the outcome at population level, which is a task under a probabilistic paradigm. Contrarily, a deterministic one is needed to predict the outcome at individual level. Unfortunately, this is a common misconception among researchers which falsely interpret a prediction model as an inference one. Respectively, this applies a multivariable instead of a multivariate model (Hidalgo and Goodman, 2013). Therefore, a multivariable model can facilitate utilization of routine health data for predicting depressive symptoms. This was to avoid questionnaire fatigue due to high frequency of screening (Egleston et al., 2011).

The objective of this study warranted a predictive modeling instead of an indicator identification. The latter analysis requires a univariate or multivariate model instead of a univariable or multivariable model which is appropriate for the former one (Hidalgo and Goodman, 2013).

References:

##### Egleston, B.L., Miller, S.M., and Meropol, N.J. (2011). The impact of misclassification due to survey response fatigue on estimation and identifiability of treatment effects. Stat Med 30, 3560-3572. 10.1002/sim.4377.

##### Hidalgo, B., and Goodman, M. (2013). Multivariate or multivariable regression? Am J Public Health 103, 39-40. 10.2105/ajph.2012.300897.

#### 2.1.5 #### The authors propose using their model as a screening tool for administering the GDS-15. In the external validation, results showed either moderate sensitivity/specificity (SPC-GBM), or high sensitivity and low specificity (DI-VNN). The authors should add a discussion of how these results impact the usage of each model where either (1) the model will incorrectly classify individuals experiencing depressive symptoms, or (2) there will be a large amount of over prediction.

Response:

Thank you this valuable suggestion. It will point out the potential impact of our model. We have added a discussion on the model usage regarding the external validation results, as shown below:

Moderate sensitivity and specificity have been demonstrated by SPC-GBM based on external validation with either local or non-local ethnicity. Among individuals experiencing depressive symptoms (i.e. positives), an incorrect prediction (i.e. a negative) may cause an individual undiagnosed. Hence, a predicted negative should be confirmed by DI-VNN which demonstrated high sensitivity by external validation. Contrarily, among individuals without depressive symptoms, this may cause overdiagnosis. However, the false positives will be screened by GDS-15, instead of being a definitive diagnosis.

#### 2.1.6 #### I found the ontology section was unclear; I am not as familiar with ontology-based methods. Could the authors add a section to their methods describing the ontology analysis, and maybe make the relationship between the ontologies, features/PCs, and their underlying meaning more clear in Figure 3?

Response:

Thank you for suggesting us to explain the ontology analysis. We have added an Ontology analysis section in the Methods, as shown below.

Ontology analysis

Our DI-VNN model could be used to explore ontological relationships among the predictors in the context of predicting the outcome. Detailed technical explanation of DI-VNN algorithm was previously described elsewhere (Sufriyana et al., 2021a). Briefly, there were three steps: (1) differential analysis for feature pre-selection; (2) structural representation of features; and (3) CNN model training.

We applied a differential analysis to choose 18 candidate features for DI-VNN among 37 predictors and 19 PCs. Differential analysis applied quantile-to-quantile normalization such that technical inter-variability (i.e. to measure predictors) was removed across the subjects. Using t-moderated statistics, a differential analysis was conducted to select candidate features. The null hypothesis was that there is no significant difference of a feature value between positives and negatives. Since a predictor could be selected by chance, which posed the analysis to multiple testing bias, we adjusted the p-values using the Benjamini-Hochberg method. We selected a feature if the adjusted p-value or false discovery rate (FDR) was less than 0.05.

After pre-selection, the candidate features without the outcome were used to construct a structural representation of feature variabilities and inter-relationships. There were two types of structural representation: (1) spatial representation; and (2) hierarchical representation. We applied t-distributed stochastic neighbor embedding (t-SNE) algorithm to spatially cluster the selected features in a three-dimensional positioning. Closer position means a higher correlation between a pair of features. To cluster the selected features in a hierarchy, we applied clique-extracted ontology algorithm. Features within the same ontology are more similar among them compared to those in the different ontology. Since these ontologies were hierarchically connected, we could evaluate which ontology was more predictive between that with less features (i.e. child ontology) and that with more features (i.e. parent ontology) after the model training.

Eventually, we used the representation as a CNN architecture and trained it using backpropagation algorithm to predict the outcome. In a CNN modeling, closer values in a multidimensional array would be summarized by the maximum value among them. By this way, inter-relationships among features were also took into account when predicting the outcome in addition to their individual values. Backpropagation algorithm in a CNN modeling also allowed to signify which features and their inter-relationships were more weighted to predict the outcome. A more-extreme weight, either positively or negatively, was represented with a higher color intensity when visualizing internal properties of our DI-VNN model. Therefore, using this ontology analysis, we could evaluate: (1) which set of features (i.e. ontology) were more predictive; (2) how these ontologies were connected; (3) what were important features in an ontology; and (4) how these features were related within an ontology.

Reference:

##### Sufriyana, H., Wu, Y.W., and Su, E.C. (2021a). Deep-insight visible neural network (DI-VNN) for improving interpretability of a non-image deep learning model by data-driven ontology, . Protocol Exchange rs.3.pex-1637/v1. 10.21203/rs.3.pex-1637/v1.

#### 2.1.7 #### The authors included a link to an online system where clinicians can upload information and the model outputs a prediction, I am assuming using the algorithm in the paper.

## 2.1.7.1 ## I am a bit worried about the public nature of this tool, given that the reliability and validity of the tool has not been published, and users could take the prediction model results at face value. I encourage the authors to add a disclaimer about the reliability/validity on the online tool at a basic reading level, so users do not take the prediction result at face value and use it for making care decisions.

Response:

Thank you for pointing this issue. We agree with reviewer’s comment that this tool could not be used before the reliability and validity are published. We have added a disclaimer regarding this issue. Please kindly check the update (https://predme.app/pre_gds15/), as shown below:

Disclaimer

This web application is intended as a prototype to showcase a tool for diagnostic prediction of depressive symptoms by GDS-15. A predicted positive is not a definitive diagnosis of depression. Do not use this web application without supervision from a competent healthcare giver.

## 2.1.7.2 ## In addition, a publication was cited online, which I am assuming is about the prediction model. The links to the publication were not working, and I could not find the publication online. See citation below: Anonymous, et al. Questionnaire-free method to predict 15-item geriatric depression scale (GDS-15) among community-dwelling elders by machine learning. EBioMedicine (2021). DOI: 00.0000/x00000-000-0000-0 PMID: 00000000 Full text PDF. Could the authors clarify if this is an existing publication, and if so, include it as a supplementary file so we can confirm that the reported results in this manuscript are different from this prior publication?

Response:

Thank you for helping us to avoid this potential misunderstanding. The reference was intended to cite this manuscript, not another publication. This is why reviewer could not find it online. We have not promoted the link of this web application anywhere yet. However, we agree with reviewer to remove the reference before this manuscript is published. Please kindly check the update (https://predme.app/pre_gds15/).

## 2.1.7.3 ## Finally, when adding my information within the online tool and looking at the GDS-15, I wondered if collecting the information used in the prediction models would really be less burdensome than taking the GDS-15 itself, which is a more direct measure of depression symptoms. In addition, I feared that the tool simply shifts the burden on reporting and entry from the patient to clinicians, who would then need to collect this information for multiple patients, and run the tool. Given this, please justify why a prediction model using these types of demographic data still has utility.

Response:

Thank you to show us the missing point. The web application was intended as a prototype to showcase our predictive system, but the future implementation should be incorporated into an HER system for automatic prediction. We revised wording related this point in the manuscript and on the web, as shown below.

Introduction

It should be incorporated to an electronic health record (EHR) system with automatic run based on pre-existing, required information in EHR.

Methods

A web application (https://predme.app/pre_gds15) is provided as a prototype, but the future implementation should be incorporated into an EHR system for automatic prediction based on pre-existing information.

Results

In addition, to deploy our models, we provided a web application (https://predme.app/pre_gds15) using the best models, as a prototype before incorporating the application into an EHR system.

Discussion

Note, our system should also be conducted in automatic manner to recommend a GDS-15 evaluation. Manual input by clinicians considerably cancels out the objective of this predictive system.

Disclaimer

Implementation of this predictive system should be incorporated to an electronic health record (EHR) system. Our system should also be conducted in automatic manner; thus, a GDS-15 evaluation is recommended by our system based on pre-existing, required information in EHR. Manual input by clinicians considerably cancels out the objective of this predictive system.

###### 2.2 ###### Minor comments

#### 2.2.1 #### The Authors state that the 15-item GDS is the “most appropriate” version of the scale. What defines “appropriate” in this context? Please clarify in the text.

Response:

Thank you for asking to clarify the context. We have revised it in the text, as shown below:

To screen depressive symptoms in older adults, the Geriatric Depression Scale (GDS) is one of the most commonly used questionnaires, of which the 15-item version (GDS-15) is the most accurate (an area under the summary receiver operating characteristic [SROC] curve of 0.900, 95% confidence interval [CI] 0.800 to 0.950) compared to either the shorter or longer versions (Krishnamoorthy et al., 2020).

Reference:

##### Krishnamoorthy, Y., Rajaa, S., and Rehman, T. (2020). Diagnostic accuracy of various forms of geriatric depression scale for screening of depression among older adults: Systematic review and meta-analysis. Arch Gerontol Geriatr 87, 104002. 10.1016/j.archger.2019.104002.

#### 2.2.2 #### I found the statement “questions asked later in the long term were shown to lead to greater misclassifications” unclear. What do the authors mean? Are they stating that there is a delay in patients with suspected depressive symptoms receiving the questionnaire? Misclassifying what specifically?

Response:

We really appreciate this suggestion since it is the main reason why our model is needed; unfortunately, we failed to convince potential future readers about our model importance, due to suboptimal wording. More explanations have been added in the text, as shown below:

Nevertheless, the screening frequency of a questionnaire should be limited, because questions asked later in the long term were shown to lead to greater misclassifications, due to untruthful or inconsistent responses to self-alleviate individual fatigue and to reduce their time in answering questions (Egleston et al., 2011). In the previous study, greater misclassification to predict a yes-or-no outcome was also found due to the effect of screening frequency; yet, it was inconclusive how many repetitions are optimum before a substantial misclassification rate was achieved. This variably depends on the questions, the subject characteristics, and the predicted outcome of interest. Since reducing the frequency of questionnaire use is reasonably beneficial, we need a triage test of GDS-15 with questionnaire-free variables. 

Reference:

##### Egleston, B.L., Miller, S.M., and Meropol, N.J. (2011). The impact of misclassification due to survey response fatigue on estimation and identifiability of treatment effects. Stat Med 30, 3560-3572. 10.1002/sim.4377.

#### 2.2.3 #### I found the paragraph of the introduction beginning with “Depression affects 264 million people globally” to cover broader material than the previous paragraph beginning with “Depressive symptoms in older adults”. It might make sense to potentially rearrange these paragraphs to begin with the global burden of depression, then highlight issues with depression questionnaires in older adults. In addition, I believe the summary statistics of depression rates in each country do not add much value to the manuscript. Perhaps the authors could shorten this sentence, or focus on statistics relevant to the specific population studied in the manuscript.

Response:

Thank you for this suggestion. We agree that description of prevalence generally from a broader to narrower region. However, the first paragraph does not describe the burden of depression in the same sense. Furthermore, we tried to describe whole reasoning of our research immediately in the earliest paragraph. This may be subjective and a matter of writing style, which are in a room for authors’ independency. With respect to this reviewer’s comment, we propose to editor for giving us a chance to keep the current order of the paragraphs.

Nonetheless, we agree to rephrase the sentences related to the country-level depression rates, but not to remove it. A wide range of prevalences due to different screening tools was pointed out in such a way to maintain accountability for persons in the domain knowledge. This is a lead to why GDS-15 is used in our study, as it is a reliable measure with a good tradeoff between the reliability and feasibility. A diagnosis based on DSM-IV-TR/DSM-V needs experts which are less available relatively in a scale of population research. Unfortunately, current wording is insufficient to describe such reasoning. We have rephrased the sentences, as shown below:

Due to different tools used for screening depression, the range of prevalences is considerably wide (Zhang et al., 2018). The prevalence rates of depression among community-dwelling older adults were reported in Sweden (7%) (Sjöberg et al., 2017), the United States (9.8%) (Brooks et al., 2018), Nigeria (52.0%) (Igbokwe et al., 2020), India (34.4%) (Pilania et al., 2019), Singapore (13%) (Feng et al., 2013), Turkey (25.2%) (Kilavuz et al., 2018), Japan (24%) (Kitagaki et al., 2020), South Korea (72.2%) (Kim and Lee, 2015), and Malaysia (59.1%) (Leong et al., 2020). Although a standard diagnosis is made using the Diagnostic and Statistical Manual of Mental Disorders V (DSM-V), the GDS-15 is quite reliable for such purposes. Prevalences in Sweden were quite similar between those based on the GDS-15 (7%) and DSM-IV-TR/DSM-V (6.6%) (Sjöberg et al., 2017).

References:

#### Brooks, J.M., Titus, A.J., Bruce, M.L., Orzechowski, N.M., Mackenzie, T.A., Bartels, S.J., and Batsis, J.A. (2018). Depression and Handgrip Strength Among U.S. Adults Aged 60 Years and Older from NHANES 2011-2014. J Nutr Health Aging 22, 938-943. 10.1007/s12603-018-1041-5.

##### Feng, L., Yap, K.B., and Ng, T.P. (2013). Depressive symptoms in older adults with chronic kidney disease: mortality, quality of life outcomes, and correlates. Am J Geriatr Psychiatry 21, 570-579. 10.1016/j.jagp.2012.12.020.

##### Igbokwe, C.C., Ejeh, V.J., Agbaje, O.S., Umoke, P.I.C., Iweama, C.N., and Ozoemena, E.L. (2020). Prevalence of loneliness and association with depressive and anxiety symptoms among retirees in Northcentral Nigeria: a cross-sectional study. BMC Geriatr 20, 153. 10.1186/s12877-020-01561-4.

##### Kilavuz, A., Meseri, R., Savas, S., Simsek, H., Sahin, S., Bicakli, D.H., Sarac, F., Uyar, M., and Akcicek, F. (2018). Association of sarcopenia with depressive symptoms and functional status among ambulatory community-dwelling elderly. Arch Gerontol Geriatr 76, 196-201. 10.1016/j.archger.2018.03.003.

##### Kim, K., and Lee, M. (2015). Depressive Symptoms of Older Adults Living Alone: The Role of Community Characteristics. Int J Aging Hum Dev 80, 248-263. 10.1177/0091415015590315.

##### Kitagaki, K., Murata, S., Tsuboi, Y., Isa, T., and Ono, R. (2020). Relationship between exercise capacity and depressive symptoms in community-dwelling older adults. Arch Gerontol Geriatr 89, 104084. 10.1016/j.archger.2020.104084.

##### Leong, O.S., Ghazali, S., Hussin, E.O.D., Lam, S.K., Japar, S., Geok, S.K., and Azmi, I.S.M. (2020). Depression among older adults in Malaysian daycare centres. Br J Community Nurs 25, 84-90. 10.12968/bjcn.2020.25.2.84.

##### Pilania, M., Yadav, V., Bairwa, M., Behera, P., Gupta, S.D., Khurana, H., Mohan, V., Baniya, G., and Poongothai, S. (2019). Prevalence of depression among the elderly (60 years and above) population in India, 1997–2016: a systematic review and meta-analysis. BMC Public Health 19, 832. 10.1186/s12889-019-7136-z.

##### Sjöberg, L., Karlsson, B., Atti, A.R., Skoog, I., Fratiglioni, L., and Wang, H.X. (2017). Prevalence of depression: Comparisons of different depression definitions in population-based samples of older adults. J Affect Disord 221, 123-131. 10.1016/j.jad.2017.06.011.

#### 2.2.4 #### The authors state in the Introduction that logistic regression is an insufficient model to develop a triage test for GDS-15 screening, but do not provide reasoning why it is insufficient. An LR model to predict GDS-15 - assuming high sensitivity, specificity, and positive predictive value - would be an ideal model to use due to its explainability, simplicity, and robustness. Please justify further why simple models are insufficient for the specific GDS-15 triage test problem.

Response:

We completely agree with reviewer on the explainability, simplicity, and robustness of LR. Generally speaking, LR demonstrates such strength; yet, assuming this holds to any real-world data is a priori conclusion. Such agreement is already stated in a probable manner. Accordingly, we also applied LR as one of the models being compared in a rigorous way (i.e. external validation with different characteristics of targeted population). Unfortunately, LR model in this study is not well-calibrated, making it ineligible for further evaluation for its discrimination ability. In the context of our study, predictive performance and robustness come first before explainability and simplicity. In addition, simplicity is not relevant for this study, since an application is provided and intended to be incorporated in EHR system for automatic run. To signify the message, we slightly changed the sentences, as shown below:

However, developing this model under a traditional approach, i.e., using a logistic regression (LR) algorithm, may be insufficient. In addition to LR, we also need other machine learning algorithms which is a field of science concerned with how machines learn from data (Deo, 2015), not limited to those based on statistical probability theory.

Reference:

Deo, R.C. (2015). Machine Learning in Medicine. Circulation 132, 1920-1930. 10.1161/circulationaha.115.001593.

#### 2.2.5 #### In the revision, per PLOS ONE’s recommendations, please include the Methods section following the Introduction, before the Results section. Thank you. See: https://journals.plos.org/plosone/s/submission-guidelines

Response:

Thank you for reminding us. We have re-arranged the section according to the guidelines.

#### 2.2.6 #### How/when was the GDS-15 delivered? During the same screening where the predictor data was collected?

Response:

We agree with reviewer that it’s not well-signified in the text. A revision to signify the data outcome/predictor collection is accordingly done, as shown below:

The trained assessor assisted a participant to fill out the GDS questionnaire. The assessor was also blinded to the predictor information. Predictor data were demographic data and routine physical health check results collected at the same time with that for GDS. The data were collected by other healthcare givers without knowing the assessment results of depressive symptoms. 

#### 2.2.7 #### In the methods, the authors state: “Meanwhile, over-diagnosis causes an increasing frequency of the use of the GDS-15, which may lead to further misclassification.” Is this true? What does “misclassification” mean in this sense? I believe that administering the GDS-15 after the prediction model would simply validate or mis-validate the prediction model results, not lead to further “misclassification” as the GDS-15 is the “gold standard” in the paper. Maybe the authors could elaborate on other issues that may arise by over-predicting patients experiencing depression symptoms.

Response:

Thank you for asking us to clarify this point. Reasonably, this may also arise the same confusion to future readers, if the main reason of the model development is not clearly described (see our response to comment 2.2.2). We have accordingly revised this sentence, as shown below:

Meanwhile, over-diagnosis causes an increasing frequency of the use of the GDS-15 for each older adult, which may lead to further misclassification, because the repetitive screening may cause response fatigue and rush which lead to higher measurement error (Egleston et al., 2011).

Reference:

##### Egleston, B.L., Miller, S.M., and Meropol, N.J. (2011). The impact of misclassification due to survey response fatigue on estimation and identifiability of treatment effects. Stat Med 30, 3560-3572. 10.1002/sim.4377.

#### 2.2.8 #### In the Methods, the authors claim when referring to RF and GBM algorithms: “Both algorithms are the most used competition-winning algorithms for predictions using tabular data.” Do the authors have a citation to back up this claim, and subsequently, why do competition-winning algorithms apply to research and this specific prediction problem? Please provide a better justification.

Response:

Thank you for notifying us about this. We apologize for missing the citation. It has been added and subsequently elaborated, as shown below:

Both algorithms are the most used competition-winning algorithms for predictions using tabular data compared to other 177 algorithms using 121 datasets (Fernandez-Delgado et al., 2014). While this is not outcome-specific, predictive modeling in a competition is independently validated; thus, the predictive performances of RF and GBM are considerably reliable and fairly evaluated.

Reference:

##### Fernandez-Delgado, M., Cernadas, E., Barro, S., and Amorim, D. (2014). Do we Need Hundreds of Classifiers to Solve Real World Classification Problems? J Mach Learn Res 15, 3133-3181. https://jmlr.org/papers/volume15/delgado14a/delgado14a.pdf.

#### 2.2.9 #### What do the authors mean when they state: “for which characteristics do not imply the data but predict the outcome very well”? Please rewrite this statement for clarity.

Response:

Thank you for this suggestion. We have rewritten the statement, as shown below:

This addresses criticisms of the CNN as a black-box model, i.e., it cannot be explained which features and how these result in a particular prediction; yet, a CNN model can predict an outcome very well.

#### 2.2.10 #### Why will the data only be available for one year after publication? Can the data be accessed now?

Response:

Thank you for asking this question. The data restriction was made because we also had used the data as a part of another study under review elsewhere. However, our analytical codes were already publicly accessed. Previously, our wording on code availability was falsely understood. Therefore, we have changed from “Data and code availability” to “Data availability statement” with the updated statement, as shown below:

All relevant data files are publicly available from the Github database (https://github.com/herdiantrisufriyana/colab_dep).

#### 2.2.11 #### The title for the first subsection of the Results states “Most had not obtained a university education, are not separated/divorced, and are religious believers”. Could the authors be more specific on who “Most” refers to?

Response:

We agree with reviewer that it is still unclear. More specific description has been given, as shown below:

Most subjects had not obtained a university education, are not separated/divorced, and are religious believers

Table 1. Most subjects had not obtained a university education, are not separated/divorced, and are religious believers.

#### 2.2.12 #### In the results, the authors state “Meanwhile, of 17 predictors and 37 PCs for the DI-VNN, only 18 of them had an FDR of <0.05 by the differential analysis with the Benjamini-Hochberg correction.” What differential analysis did the authors perform? What were the null and alternative hypotheses? Please state in the main text.

Response:

Thank you for pointing out the differential analysis description. We have added more explanation in the Methods (a part of our response to comment 2.1.6), as show below:

We applied a differential analysis to choose 18 candidate features for DI-VNN among 37 predictors and 19 PCs. Differential analysis applied quantile-to-quantile normalization such that technical inter-variability (i.e. to measure predictors) was removed across the subjects. Using t-moderated statistics, a differential analysis was conducted to select candidate features. The null hypothesis was that there is no significant difference of a feature value between positives and negatives. Since a predictor could be selected by chance, which posed the analysis to multiple testing bias, we adjusted the p-values using the Benjamini-Hochberg method. We selected a feature if the adjusted p-value or false discovery rate (FDR) was less than 0.05.

#### 2.2.13 #### I would appreciate if the authors stated the AUROC of the best performing models for the external validation in the Results section of the main text. I realize it is on Table 2.

Response:

Thank you for this suggestion because it will improve the readability of our manuscript. We have stated the AUROCs, as shown below:

Predictive performances of the SPC-GBM (AUROC of 0.578, 95% CI 0.572 to 0.583; n=250) were shown to be similar to those of the DI-VNN without re-calibration (AUROC of 0.577, 95% CI 0.576 to 0.579; n=250) in an external validation set with a local ethnic group. In addition, the DI-VNN also showed similar predictive performances among those using training (AUROC of 0.577, 95% CI 0.576 to 0.579; n=1002) and two test sets, either with a local (AUROC of 0.577, 95% CI 0.576 to 0.579; n=250) or non-local (AUROC of 0.577, 95% CI 0.576 to 0.579; n=129) ethnic group.

#### 2.2.14 #### What methods were used to identify the important features from the SPC-GBM and DI-VNN models? I know that it is often difficult to extract important features in deep learning algorithms. Thus, I would be interested in how the authors identified the important features in the algorithm.

Response:

Thank you for asking this question. Additionally, we explored the important features after achieving our main objective to clarify the accuracy of utilizing routine data for a triage test of GDS-15. This was already explained elsewhere. The papers are cited in current revision, as shown below:

Methods

In addition, we explored the best model to identify important features post-analysis (see Results).

Results

We could identify how important the predictors are in predicting the GDS-15 using both models. There were seven PCs in the SPC-GBM that represent latent variables with different weights among them on the 37 predictors. Details are described elsewhere on how the weights were inferred (Sufriyana et al., 2021b). We visualized absolute values of these weights for each selected PC (Figure 2). Absolute values were used because the positive/negative values cannot be interpreted in a straightforward manner regardless of whether these tend to be events or non-events. By observing the visualization, we could infer the meaning of the latent variables. These were named based on the higher absolute values by referring to particular predictors.

While PCs in the SPC-GBM were independently interpreted, those could be interconnected in the DI-VNN (Figure 3), including the PCs and the predictors of origin. Each ontology predicted an outcome in the DI-VNN, contributing to optimization of the predictive performance. If we used the model architecture up to each ontology for predicting the outcome, different AUROCs were shown (Figure 3a). The top three highest AUROCs were those predicted up to the root, ONT:20, and ONT:22. Each ontology was visualized for the array difference between GDS-15 positives and negatives (Figure 2b). The weighted features for GDS-15 positives is subtracted by those for the negatives. This means positive and negative results from this subtraction respectively referred to GDS-15 positive and negative predictions. Details are described elsewhere on how each ontology individual prediction was taken into the final prediction and which layers were used for feature visualization (Sufriyana et al., 2021a).

References:

##### Sufriyana, H., Wu, Y.W., and Su, E.C. (2021a). Deep-insight visible neural network (DI-VNN) for improving interpretability of a non-image deep learning model by data-driven ontology. Protocol Exchange rs.3.pex-1637/v1. 10.21203/rs.3.pex-1637/v1.

##### Sufriyana, H., Wu, Y.W., and Su, E.C.Y. (2021b). Resampled dimensional reduction for feature representation in machine learning. Protocol Exchange rs.3.pex-1636/v1. 10.21203/rs.3.pex-1636/v1.

#### 2.2.15 #### Why do the authors believe there was such significant overfitting in the SPC-GBM model from the internal validation to the external validation cohorts?

Response:

Thank you for asking us to explain overfitting of SPC-GBM. We have elaborated our reason for the overfitting, as shown below.

Both the RF and GBM algorithms achieved suitable predictive performances by overfitting the training set, as observed in this study. For example, predictive performances of SPC-GBM were reduced by 42.08% and 37.98% respectively for point estimates of AUROCs using validation sets with local (0.578, 95% CI 0.572 to 0.583) and non-local (0.619, 95% CI 0.610 to 0.627) ethnicities, compared to those using training set (0.998, 95% CI 0.998 to 0.998).

 

################################################################################

######## 3 ######## Reviewer 2

################################################################################

###### 3.1 ###### Key strength of the paper

#### 3.1.1 #### The work is important, instruments that could potentially be used to screen for these kinds of conditions without the immediate input of a patient is highly relevant and an important facet of medical technology

Response: 

Thank you for your appreciation. Our intention in developing the model is well-described by reviewer. However, we admit that the readability of this manuscript should be improved furthermore.

###### 3.2 ###### Main weakness of the paper

#### 3.2.1 #### The methodology of this study is very poorly done, or poorly presented. The authors give numbers associated with the models’ results, but the metrics they choose to present are not very meaningful given the context, and furthermore the authors spend almost no time explaining exactly what types of inputs or how hyper-parameters or even how data-splitting occurred. The paper is not written very strongly, and there are many details from the implementation and data preparation that are missing.

Response:

Thank you. It matters for us to receive these critics to improve our manuscript. Since our work focus on a novelty in the domain field, we wrote this manuscript with clinical audience in mind. However, it is fully understood, because of multidisciplinary nature of this study, which was also the reason of why PLOS One, writing the manuscript is challenging in order to be readable for both clinical and computational audiences. Completeness for one may be considerably distracting for another. Meanwhile, a word count limit should also be followed. Although we have briefly described hyper-parameter tuning method, data splitting, and data preparation (preprocessing), as shown below, we agree to believe these are insufficient (i.e. “almost no time explaining”). This is not to mention we have made our analytical codes publicly-accessible (https://github.com/herdiantrisufriyana/colab_dep). Since it is still unclear why “the metrics they choose to present are not very meaningful given the context”, we would follow the next comments, which are possibly related, as a clear elaboration from reviewer for this concern. We have added more descriptions on hyper-parameter tuning method, data splitting, and data preparation (preprocessing), as shown below:

An elastic net regression algorithm was applied in which L1- and L2-norm regularization was conducted. We chose this regularization method over others to minimize both of the excluded predictors and to prevent overfitting (Moons et al., 2019). Hyperparameter tuning was conducted by a random search with up to 10 configurations of alpha and lambda values as L1- and L2-norm regularization factors, respectively.

Hyperparameter tuning was also conducted by a random search over six configurations of the number of predictors being sampled a time for RF and number of trees, maximum depth of a tree, and shrinkage factor for the GBM. Minimum samples per node were also configured for both models.

Details about the DI-VNN pipeline were previously described elsewhere (Sufriyana et al., 2021a). Some modifications of this pipeline were those by applying this procedure over 37 predictors and 19 PCs resulting 18 candidate features for DI-VNN. These were centered using each average value after quantile-to-quantile normalization over all features among samples.

Data partitioning was conducted to obtain both internal and external validation sets. We used participants with ethnicity not from Sulawesi Island for the external validation set.

We also randomly split the remaining set after excluding the external validation set. This provided another external validation set with as much as ~20% of the remaining set. For the first to third models, we applied 10-fold cross-validation for hyperparameter tuning and 30 times bootstrapping for training the model using the best hyperparameters. We also applied 10-fold cross-validation to compute the rotated matrix of PCs. For the fourth model, we applied a hold-out cross-validation with 80:20 ratios for the training and validation sets.

References:

##### Moons, K.G.M., Wolff, R.F., Riley, R.D., Whiting, P.F., Westwood, M., Collins, G.S., Reitsma, J.B., Kleijnen, J., and Mallett, S. (2019). PROBAST: A Tool to Assess Risk of Bias and Applicability of Prediction Model Studies: Explanation and Elaboration. Ann Intern Med 170, W1-w33. 10.7326/m18-1377.

##### Sufriyana, H., Wu, Y.W., and Su, E.C. (2021a). Deep-insight visible neural network (DI-VNN) for improving interpretability of a non-image deep learning model by data-driven ontology. Protocol Exchange rs.3.pex-1637/v1. 10.21203/rs.3.pex-1637/v1.

#### 3.2.2 #### Why did the authors not utilize a cross-fold validation approach?

Response:

Thank you for this suggestion. We have utilized a cross-fold validation in our analytical pipeline, as shown below. Within the scope of this comment, we do not find a way to elaborate it furthermore.

For the first to third models, we applied 10-fold cross-validation for hyperparameter tuning and 30 times bootstrapping for training the model using the best hyperparameters. We also applied 10-fold cross-validation to compute the rotated matrix of PCs. For the fourth model, we applied a hold-out cross-validation with 80:20 ratios for the training and validation sets. 

#### 3.2.3 #### What does internally/externally validated data mean in the context of splitting the data for training and testing?

Response:

Thank you for pointing out these terms. We have added an explanation what internal and external validation sets refer to, as shown below:

Data partitioning was conducted to obtain both internal and external validation sets. Respectively, this meant we have a training set and two test sets.

#### 3.2.4 #### Neural Networks typically require orders of magnitude more data to justify over conventional models such as Decision Trees or Support Vector Machines, why not use those?

Response:

We appreciate this comment which may gain more attentions from future readers on the comparison we made. It applied both conventional algorithms and ones we developed using neural network. We reinforced this message, as shown below:

Although there are abundant machine learning algorithms for a model development, no exhaustive comparison was made over all the available algorithms. This because a larger number of model in comparison would be more vulnerable to a multiple-testing effect in relative to the number of datasets, i.e., the best model is found to be well-performed using a dataset with a particular partitioning simply by chance (Westphal et al., 2022).

Sufficient sample size was also considered according to the PROBAST guidelines, since a small sample size was considerably vulnerable to overfitting (Moons et al., 2019). The three types of algorithms also covered those with the lowest and highest sample size requirements, which were 20 (i.e. logistic regression) and >200 (i.e., random forest [RF] and neural network) events per variable (EPVs), according to a previous study (van der Ploeg et al., 2014). This also identified 50 and >200 EPVs respectively for decision tree and support vector machine. We did not use both which neither commonly outperforming other algorithms nor requiring a sample size small enough for this study. Although we used algorithms which require >200 EPVs, we evaluated the models using a rigorous data splitting which would identify overfitting by comparing the evaluation results between internal and external validation sets with same and different characteristics for a particular circumstance (see Model validation), as recommended by the PROBAST guidelines (Moons et al., 2019).

References:

##### Moons, K.G.M., Wolff, R.F., Riley, R.D., Whiting, P.F., Westwood, M., Collins, G.S., Reitsma, J.B., Kleijnen, J., and Mallett, S. (2019). PROBAST: A Tool to Assess Risk of Bias and Applicability of Prediction Model Studies: Explanation and Elaboration. Ann Intern Med 170, W1-w33. 10.7326/m18-1377.

##### van der Ploeg, T., Austin, P.C., and Steyerberg, E.W. (2014). Modern modelling techniques are data hungry: a simulation study for predicting dichotomous endpoints. BMC Med Res Methodol 14, 137. 10.1186/1471-2288-14-137.

##### Westphal, M., Zapf, A., and Brannath, W. (2022). A multiple testing framework for diagnostic accuracy studies with co-primary endpoints. Stat Med 41, 891-909. 10.1002/sim.9308.

#### 3.2.5 #### All of these models are very sensitive to the hyper-parameters you choose, and the types of data you pass in, which were chosen, what was the justification behind it?

Response:

Thank you for this suggestion. We have added the justification for the types of data passed in the models and the chosen hyperparameters, as shown below:

To avoid such comparison, we considered three criteria to choose algorithms for developing the models: (1) those commonly used in clinical prediction studies, i.e., logistic regression (Moons et al., 2019), which expects a linear predictor-outcome correlation; (2) those which commonly outperformed others (177 algorithms) across 121 datasets (Fernandez-Delgado et al., 2014), which allow a non-linear predictor-outcome correlation; and (3) our proposed neural-network algorithm (Sufriyana et al., 2021a), which pursues moderate predictive performance and deeper interpretability.

In addition, for hyperparameter tuning, we used random-search method to determine values for the pre-defined hyperparameters, or those which were defined before conducting this study in a pre-registered protocol (Sufriyana et al., 2021a). The randomness and pre-registration were intended to avoid a research bias so called “hypothesizing after the results are known (HARking)” (Rubin, 2017). In this study, it is a situation in which a set of hyperparameters for an algorithm, as a hypothesis, is specifically defined to achieve the only acceptable predictive performance in an external validation set which should be used for any parts of model selection.

By L1- and L2-norm regularization, we set the hyperparameter tuning process to tradeoff between removing and maintaining the number of predictors used for predicting the outcome; thus, we could infer which variables have predictive values under a simple predictive modeling framework.

We defined these hyperparameter variables in aggregate between both the tree-based ensemble learners to pursue a wide range of configurations. For example, we applied a different number of predictors being sampled a time for RF while maintaining the same tree structure, but, contrarily, we applied different tree structures for GBM while maintaining the same number of predictors sampled a time. The best hyperparameters were selected for each of the algorithms under a variety of samples per node to take into account the effect of sampling error. Therefore, we expected hypothesis search of hyperparameters well-covered while avoiding the pitfall of HARking.

To avoid HARking, we followed the same hyperparameter tuning approach which already pre-registered and completely described elsewhere (Sufriyana et al., 2021a).

References:

##### Moons, K.G.M., Wolff, R.F., Riley, R.D., Whiting, P.F., Westwood, M., Collins, G.S., Reitsma, J.B., Kleijnen, J., and Mallett, S. (2019). PROBAST: A Tool to Assess Risk of Bias and Applicability of Prediction Model Studies: Explanation and Elaboration. Ann Intern Med 170, W1-w33. 10.7326/m18-1377.

##### Fernandez-Delgado, M., Cernadas, E., Barro, S., and Amorim, D. (2014). Do we Need Hundreds of Classifiers to Solve Real World Classification Problems? J Mach Learn Res 15, 3133-3181. https://jmlr.org/papers/volume15/delgado14a/delgado14a.pdf.

##### Rubin, M. (2017). When does HARKing hurt? Identifying when different types of undisclosed post hoc hypothesizing harm scientific progress. Review of General Psychology 21, 308-320. 10.1037/gpr0000128.

##### Sufriyana, H., Wu, Y.W., and Su, E.C. (2021a). Deep-insight visible neural network (DI-VNN) for improving interpretability of a non-image deep learning model by data-driven ontology. Protocol Exchange rs.3.pex-1637/v1. 10.21203/rs.3.pex-1637/v1.

#### 3.2.6 #### While the results are generally reported, no meaningful discussion about baselines are provided, how much better than random guessing are these models performing?

Response:

Thank you for making us notice on this important viewpoint. We have revised the manuscript in the Methods, Results, and Discussion related to this issue, as shown below:

Methods

The best model was determined using the internal validation set, and should be robust based on all external validation sets, and for which the central value of the AUROC was approximately >0.5, as a baseline value to determine if a predictive performance of a model was better than a random or coin-flip guessing.

Results

In addition, according to any metrics evaluated in this study, predictive performances of SPC-GBM were better than a random or coin-flip guessing (e.g. AUROC point estimate of SPC-GBM >0.5).

Discussion

Nonetheless, using the baseline value, the predictive performance of SPC-GBM was better than a random or coin-flip guessing for any metrics evaluated in this study.

#### 3.2.7 #### Why is AUROC the chosen metric, and not F1-score or precision and recall?

Response:

Thank you for pointing this out. We did not only use AUROC, and both precision and recall were used. To signify both metrics for future readers, we have added the terms adjacent to those in the manuscript, as shown below:

Comparing to the same baseline value, we also computed the specificity, accuracy, positive predictive value (PPV) or precision, and negative predictive value (NPV) using a threshold at approximately a sensitivity or recall of ~90% or a false negative rate of ~10%, because the risk of under-diagnosis outweighs that of over-diagnosis.

#### 3.2.8 #### These questions are just a few of the many that are left to the reader to try to discern or figure out potentially by having access to the data. While this does not mean that the results presented are not valid ones, there is no understanding of how good they really are? (Would a loaded coin-flip perform better?)

Response:

Thank you for elaborating this concern furthermore. However, we disagree that having access is the only way for the future readers to discern detailed, technical procedures of cross-fold validation, data splitting, hyperparameter tuning, and model evaluation, as pointed by reviewer in the previous comments. This reproducibility goal can be achieved by sharing the analytical codes (https://github.com/herdiantrisufriyana/colab_dep) with detailed description (comments), which were already done. Regarding the data accessibility, it is a common practice for a healthcare dataset to limit its access due to local policy by requesting it to the corresponding author. We have added more discussion related to how good the models are (see comment 3.2.6). No further change is made for this comment since there is no other specific questions.

###### 3.3 ###### Novelty/originality

#### 3.3.1 #### Taking into account the relevance of the work for the PLOS ONE audience.

Response:

We appreciate reviewer for suggesting further elaboration of the relevance to the PLOS ONE audience. This manuscript was mainly written with clinical audience in focus. This is included in that covered by the PLOS ONE. However, due multidisciplinary nature of this study, we understand the challenges for finding the peer reviewers, which we appreciate for the editor in this journal. According to this context, we have clarified the relevance our work, as shown below:

To screen depressive symptoms in older adults, the Geriatric Depression Scale (GDS) is one of the most commonly used questionnaires, of which the 15-item version (GDS-15) is the most accurate (an area under the summary receiver operating characteristic [SROC] curve of 0.900, 95% confidence interval [CI] 0.800 to 0.950) compared to either the shorter or longer versions (Krishnamoorthy et al., 2020). Nevertheless, the screening frequency of a questionnaire should be limited, because questions asked later in the long term were shown to lead to greater misclassifications, due to untruthful or inconsistent responses to self-alleviate individual fatigue and to reduce their time in answering questions (Egleston et al., 2011). In the previous study, greater misclassification to predict a yes-or-no outcome was also found due to the effect of screening frequency; yet, it was inconclusive how many repetitions are optimum before a substantial misclassification rate was achieved. This variably depends on the questions, the subject characteristics, and the predicted outcome of interest. Since reducing the frequency of questionnaire use is reasonably beneficial, we need a triage test of GDS-15 with questionnaire-free variables. Demographic and physical health data from routine visits can be utilized. This is because older adults with depressive symptoms may present with more physical complaints, implying a psychological change that might be overlooked by caregivers (Kok and Reynolds, 2017). However, the accuracy of utilizing such data for a triage test is still unclear.

References:

##### Egleston, B.L., Miller, S.M., and Meropol, N.J. (2011). The impact of misclassification due to survey response fatigue on estimation and identifiability of treatment effects. Stat Med 30, 3560-3572. 10.1002/sim.4377.

#### Kok, R.M., and Reynolds, C.F., 3rd (2017). Management of Depression in Older Adults: A Review. Jama 317, 2114-2122. 10.1001/jama.2017.5706.

##### Krishnamoorthy, Y., Rajaa, S., and Rehman, T. (2020). Diagnostic accuracy of various forms of geriatric depression scale for screening of depression among older adults: Systematic review and meta-analysis. Arch Gerontol Geriatr 87, 104002. 10.1016/j.archger.2019.104002.

#### 3.3.2 #### While the domain might be novel, the general approach of using machine learning models in this way is not particularly novel.

Response:

Thank you for this comment. Reviewer have captured the novelty of this study, as we intended. Since the methodological novelties of several methods in this study were already described elsewhere, we described the novelty of this study in the domain knowledge for the manuscript submitted the PLOS ONE. We appreciate this journal significantly encourage more multidisciplinary studies; otherwise, an application of scientific discovery, which is often multidisciplinary, would be reported in a very limited venue. No change is made for this comment.

###### 3.4 ###### Technical/theoretical Correctness

#### 3.4.1 #### Taking into account datasets, baselines, experimental design, affective theory, are there enough details provided to be able to reproduce the experiments and understand the contribution?

Response:

Thank you for the critics. We assume that this comment wraps up the previous ones we have addressed. Please kindly see our responses to: (1) comment 3.2.8 for account datasets; (2) comments 3.2.6 and 3.2.7 for baselines; and (3) comments 3.2.1 to 3.2.5 for experimental design. It is unclear what reviewer meant by “affective theory”. Yet, broadly interpreted, we addressed this as a lack of readability for this manuscript before the previous comments were addressed.

#### 3.4.2 #### There is a plethora of missing details that are unjustified and under-reported making the correctness of this work difficult to evaluate and exact reproducibility difficult.

Response:

We agree with reviewers that several parts of this manuscript were not well-justified and well-reported. For the reproducibility, this is true to some extent, as our response to comment 3.2.8. We also assume that this comment wraps up the previous ones we have addressed.

###### 3.5 ###### Quality of references

#### 3.5.1 #### Is it a good mix of older and newer papers? Do the authors show a good grasp of the current state of the literature? Do they also cite other papers apart from their own work?

Response:

Thank you for pointing this out. Generally speaking, we agree with reviewer that a quantity of newer papers shows a better grasp of the current state of the literature. Yet, it is highly contextual. First, of 74 papers cited in this manuscript, most (n=46/74, 62.16%) were newer papers published from 2017 to current year. In this study, we intended to conduct a multivariable prediction model, instead of a predictor finding study. Please kindly consult the PROBAST guidelines for the difference (Wolff et al., 2019). Since this study was intended to solve the questionnaire fatigue problem using the existing routine data (see comment 3.5.2.1), we started from the existing variables in the routine data to search in the literature for predictor finding studies that support predictive capability of the existing variables. This approach resulted in some of the older papers, i.e., 2016 (n=3/5, 60%), 2015 (n=1/6, 17%), 2014 (n=1/4, 25%), 2012 (n=1/1, 100%), 2011 (n=3/4, 75%), and 2010 (n=1/1, 100%) papers. This accounted for 35.71% of the older papers (n=10/28). The remaining older papers were standard guidelines and questionnaires (n=6/18, 33.3%). Therefore, the references were reasonably cited.

We also cited other papers apart from our work in majority (n=69/74, 93.24%). Of 5 papers from our own work, there were 2 protocol papers (40%). It is more appropriate and currently recommended to cite a protocol paper instead of describing the analytical procedures, redundantly. In the previous version of this manuscript, we only cited lesser number of the papers to avoid self-citation. Since more details were requested by reviewers, we need to cite more of our protocol papers. Hence, it is inevitable for an applied research that requires pre-registered, complex analyses. In addition, the remaining papers were cited as a part of wide application examples of machine learning in medicine (n=1/6, 16.7%), and as an example of under-appreciated medical history usage in EHR (beyond sophisticated data) for predicting a disease. The latter reason was directly related to this study.

No change was made in the manuscript, related to this comment.

#### 3.5.2 #### The chosen references seem to be reasonable but the way is utilized throughout the paper is not very good. Many things the authors claim go unjustified or uncited, or cited awkwardly. e.g.:

## 3.5.2.1 ## “Nevertheless, the screening frequency of a questionnaire should be limited, because questions asked later in the long term were shown to lead to greater misclassifications (Egleston et al., 2011).” This sentence doesn't add anything and is unclear. What are "questions asked later in the long term", and how exactly do they lead to greater misclassification? What is the screening frequency that is most appropriate? Does it vary by the depressive state of the individual?

Response:

We really appreciate this suggestion since it is the main reason why our model is needed; unfortunately, we failed to convince reviewer or potential future readers about our model importance, due to suboptimal wording. More explanations have been added in the text, as shown below:

Nevertheless, the screening frequency of a questionnaire should be limited, because questions asked later in the long term were shown to lead to greater misclassifications, due to untruthful or inconsistent responses to self-alleviate individual fatigue and to reduce their time in answering questions (Egleston et al., 2011). In the previous study, greater misclassification to predict a yes-or-no outcome was also found due to the effect of screening frequency; yet, it was inconclusive how many repetitions are optimum before a substantial misclassification rate was achieved. This variably depends on the questions, the subject characteristics, and the predicted outcome of interest. Since reducing the frequency of questionnaire use is reasonably beneficial, we need a triage test of GDS-15 with questionnaire-free variables.

Reference:

##### Egleston, B.L., Miller, S.M., and Meropol, N.J. (2011). The impact of misclassification due to survey response fatigue on estimation and identifiability of treatment effects. Stat Med 30, 3560-3572. 10.1002/sim.4377.

## 3.5.2.2 ## “A triage test with questionnaire-free variables is needed to reduce the frequency of questionnaire use. Demographic and physical health data from routine visits can be utilized.” This is uncited. Also, what does it mean that it is "needed", wouldn’t a redesigned questionnaire, with repeated evaluation in mind, perform better than one that wasn’t meant to (GDS-15). What about the huge population of people who don’t have access to regular medical attention and physical evaluation, or who might not have access to a good health record/history due to the unavailability of infrastructure for this purpose? Most importantly, none of this is later addressed by references or the methods presented in this work. While it might be corrected by instead offering the perspective of: “It would allow physicians a more readily and less invasive approach to screen patients for depression risk”

Response:

Thank you for elaborating this issue. We believe reviewer’s comment of “none of this is later addressed by references or the methods presented in this work” was due to misunderstanding caused by our suboptimal wording, as we responded for in comment 3.5.2.1. By current revision, the sentences quoted by reviewer should be clearly viewed as our opinion to answer why our model is needed, which did not refer to any previous studies; thus, there is no need for a citation. It is true if reviewer thought that our study did not intend to redesign questionnaire. However, our goal was neither to “allow physicians a more readily and less invasive approach to screen patients for depression risk”. In the domain field, it was already achieved by GDS-15 itself. Instead, our goal in this study was to reduce GDS-15 repetition in each older adult, since community-dwelling older adults routinely visit a healthcare facility within a year. The reduction was proposed to be done by maximizing utilization of electronic health record (EHR) from routine visits; thus, GDS-15 would be only conducted for an older adult if they are predicted positive by our models. We agree that some healthcare settings do not have a readily EHR and it is important to solve that problem, too. However, this is beyond the scope of this study, since we did not intend to find a substitute of the screening method for that in low-resource settings. Changes related to this comment are already covered by those for comment 3.5.2.1.

###### 3.6 ###### Clarity of presentation

#### 3.6.1 #### The English does not need to be flawless, but the text should be understandable.

Response:

We appreciate reviewer was willing to continue help us for improving this manuscript, although the English was not fully understandable. Since reviewer did not give specific elaboration regarding this concern, we generally revised language aspect of this paper on parts commented previously, which implied not understandable by reviewers.

#### 3.6.2 #### The presentation of this work is incredibly sub-par. Not only are there a tremendous number of sentences that don’t make sense for logical or grammatical reasons, the discussion of the work at hand is insufficient. It is difficult to follow and the paper repeats itself without clarifying or adding to previous made assertions and conclusions. Overall this is not a well written paper.

Response:

Thank you for this comment. Since there are no specific parts of the manuscript pointed by reviewer for this concern, we assume that the sentences that don’t make sense and the insufficient discussion were those commented previously. Therefore, by current revision for the previous comments, we have also addressed this comment.

---

## [Decision Letter · Decision Letter 1]

24 Nov 2022

PONE-D-22-10670R1Questionnaire-free machine-learning method to predict depressive symptoms among community-dwelling older adultsPLOS ONE

Dear Dr. Chuang,

Thank you for submitting your manuscript to PLOS ONE. After careful consideration, we feel that it has merit but does not fully meet PLOS ONE’s publication criteria as it currently stands. Therefore, we invite you to submit a revised version of the manuscript that addresses the points raised during the review process.

ACADEMIC EDITOR:Based on the reviewers' comments, you are asked to provide a revised version of the manuscript addressing all their concerns.

We look forward to receiving your revised manuscript.

Kind regards,

Tarik A. Rashid, PhD

Academic Editor

PLOS ONE

Reviewers' comments:

Reviewer's Responses to Questions

**Comments to the Author**

1. If the authors have adequately addressed your comments raised in a previous round of review and you feel that this manuscript is now acceptable for publication, you may indicate that here to bypass the “Comments to the Author” section, enter your conflict of interest statement in the “Confidential to Editor” section, and submit your "Accept" recommendation.

Reviewer #1: (No Response)

Reviewer #3: (No Response)

2. Is the manuscript technically sound, and do the data support the conclusions?

Reviewer #1: Partly

Reviewer #3: Partly

3. Has the statistical analysis been performed appropriately and rigorously? 

Reviewer #1: Yes

Reviewer #3: N/A

4. Have the authors made all data underlying the findings in their manuscript fully available?

Reviewer #1: Yes

Reviewer #3: Yes

5. Is the manuscript presented in an intelligible fashion and written in standard English?

Reviewer #1: No

Reviewer #3: Yes

6. Review Comments to the Author

Reviewer #1: Thank you to the authors for responding to my comments. Many of the edits were very helpful, but I have further recommendations that I believe could strengthen the manuscript. I also recommend having the manuscript revised for language clarity prior to the next submission. I’ve pointed out some specific examples regarding language clarity in my comments, but it would be difficult to list all of them. I recommend taking another general pass, or potentially asking an external scientific writer to help revise the manuscript. Some of these language issues are grammatical (eg, see my updated comment 2.1.2), but additionally about the specific word choices that justify the conclusions based upon the results or extracted from cited references (eg, see my updated comment 2.2.1).

My points are numbered by the numbering used by the authors in their point-by-point responses. Thank you.

2.1.2: I thank the authors for confirming that data leakage did not occur. I think the language could be made clearer. I believe the authors mean by “data partitioning” that within each cross-validation split, all standardization and MICE parameters were computed within the training data, and applied to both the training and validation data. Please confirm, and if true, clean up the language in the methods. “Data partitioned for model development” for example, is clearer language with the appropriate tense compared to “data partitioning for model development”.

2.1.3: Thank you for clarifying the intention of the external validation set in the paper. I recommend adding an extra sentence in the Limitations section stating that despite the reliability of the model demonstrated in the paper using external validation, one should still not assume generalizability. More testing is needed.

2.1.4: I understand the clarification between prediction models and risk identification. I am also not sure of the relevance of multivariate versus multivariate models the authors are making in the paper, and would recommend its removal.

On the former point regarding survey fatigue, I am not sure I agree that the authors proposed system is less arduous than depression screening. The authors’ proposed system requires the collection of clinical data (eg, on commorbidities, health conditions, hearing problems), which would require some amount of interaction with the healthcare system. One could argue that accessing annual clinician check-ins is as arduous as self-reported surveys, or a clinical depression assessment.

I also am not sure the point on “untruthful or inconsistent responses” in the Introduction is a good argument, given you’re still validating a model against a yes/no self-report of depression symptoms - something that by the authors' argument could be an “untruthful” assessment as ground truth - and thus the model would just propagate this response bias on a large scale. This is analogous to how prediction models propagate bias in data broadly, for example see, https://doi.org/10.1073/pnas.2204529119. I would suggest removing this point as well.

That being said, I think the point in the introduction on the developed model acting as an EHR indicator to triage patients for a mental health follow-up is sufficient for justifying the research, and the discussion paragraph regarding “Notably, we conducted this study [...]” can be removed.

2.2.1: I recommend changing the language to not state so objectively the accuracy of the GDS-15, but instead state that a “of which a recent systematic review and meta-analysis found that the 15-item version (GDS-15) is the most accurate”, since the authors are basing this point on a single paper.

2.2.2: I recommend removing the statement “questions asked later in the long-term”, and as suggested in my response to 2.1.4, I do not think response bias is a good motivation for this prediction model. I recommend removing this section.

2.2.4: I understand the authors’ point regarding the preference for machine learning. I recommend stating it explicitly upfront in the introduction, clarifying the word "insufficient" within the same sentence. For example: “[...] using logistic regression (LR) may be insufficient because its simplicity does not accurately reflect the complexity of real-world data [citation justifying this conclusion]. Machine learning [...]”.

Reviewer #3: I can see that authors have incorporated most of the changes as asked by the reviewers.

1. Novelty of the proposed approach is either missing or the authors fail to express it in the manuscript. Novelty should be explained.

2. Authors should explain the contributions made in the paper by adding a sub-section in the first section along with a sub-section on motivation.

3. Paper requires a flowchart which can show the flow and step by step approach of the proposed methodology in tackling the undertaken problem.

4. Formatting mistakes are there; such as random forest [RF] , here it should be written as random forest (RF), Similar errors should be removed.

7. PLOS authors have the option to publish the peer review history of their article (what does this mean?). If published, this will include your full peer review and any attached files.

Reviewer #1: No

Reviewer #3: No

---

## [Author Response · Author response to Decision Letter 1]

19 Dec 2022

################################################################################

######## 1 ######## Academic editor

################################################################################

###### 1.1 ###### Based on the reviewers' comments, you are asked to provide a revised version of the manuscript addressing all their concerns.

Response:

Thank you for giving us one more chance to revise the manuscript. We have addressed all the reviewers’ comments. The revision is tracked in the manuscript and some are either commented in the manuscript or shown in this rebuttal letter.

 

################################################################################

######## 2 ######## Reviewer 1

################################################################################

Thank you to the authors for responding to my comments. Many of the edits were very helpful, but I have further recommendations that I believe could strengthen the manuscript. I also recommend having the manuscript revised for language clarity prior to the next submission. I’ve pointed out some specific examples regarding language clarity in my comments, but it would be difficult to list all of them. I recommend taking another general pass, or potentially asking an external scientific writer to help revise the manuscript. Some of these language issues are grammatical (eg, see my updated comment 2.1.2), but additionally about the specific word choices that justify the conclusions based upon the results or extracted from cited references (eg, see my updated comment 2.2.1).

My points are numbered by the numbering used by the authors in their point-by-point responses. Thank you.

Response:

Thank you for appreciating our previous revision. We agree to take another general pass with an external writing assistance. The revision is tracked all over the manuscript.

###### 2.1 ###### Major comments

#### 2.1.2 #### I thank the authors for confirming that data leakage did not occur. I think the language could be made clearer. I believe the authors mean by “data partitioning” that within each cross-validation split, all standardization and MICE parameters were computed within the training data, and applied to both the training and validation data. Please confirm, and if true, clean up the language in the methods. “Data partitioned for model development” for example, is clearer language with the appropriate tense compared to “data partitioning for model development”.

Response:

Thank you for making the language clearer. What we mean by “data partitioning” was a subset after partitioning data for either development or validation. We agree that the language was unclear; thus, changes are made according to your suggestion. The revision is tracked and commented in the manuscript but not shown below.

#### 2.1.3 #### Thank you for clarifying the intention of the external validation set in the paper. I recommend adding an extra sentence in the Limitations section stating that despite the reliability of the model demonstrated in the paper using external validation, one should still not assume generalizability. More testing is needed.

Response:

Thank you for this suggestion. This statement is correct to some extent. We agree to add it in the limitations. The revision is tracked and commented in the manuscript, and also shown below.

Revision:

Eventually, despite the model’s reliability demonstrated in the paper using external validation, one should still refrain from assuming generalizability for any other population with different characteristics. External validation is still required for such a population, yet, this is a general issue in prediction studies, not limited to our study.

#### 2.1.4 #### I understand the clarification between prediction models and risk identification. I am also not sure of the relevance of multivariate versus multivariate models the authors are making in the paper, and would recommend its removal. On the former point regarding survey fatigue, I am not sure I agree that the authors proposed system is less arduous than depression screening. The authors’ proposed system requires the collection of clinical data (eg, on commorbidities, health conditions, hearing problems), which would require some amount of interaction with the healthcare system. One could argue that accessing annual clinician check-ins is as arduous as self-reported surveys, or a clinical depression assessment. I also am not sure the point on “untruthful or inconsistent responses” in the Introduction is a good argument, given you’re still validating a model against a yes/no self-report of depression symptoms - something that by the authors' argument could be an “untruthful” assessment as ground truth - and thus the model would just propagate this response bias on a large scale. This is analogous to how prediction models propagate bias in data broadly, for example see, https://doi.org/10.1073/pnas.2204529119. I would suggest removing this point as well. That being said, I think the point in the introduction on the developed model acting as an EHR indicator to triage patients for a mental health follow-up is sufficient for justifying the research, and the discussion paragraph regarding “Notably, we conducted this study [...]” can be removed.

Response:

Thank you for your clarification to the previous comment for this issue. We agree to remove this paragraph under section Discussion; thus, neither tracked changes nor comment are shown in the manuscript. Also thank you for appreciating our point in the Introduction. We also changed it according to comment 2.2.2 below.

###### 2.2 ###### Minor comments

#### 2.2.1 #### I recommend changing the language to not state so objectively the accuracy of the GDS-15, but instead state that a “of which a recent systematic review and meta-analysis found that the 15-item version (GDS-15) is the most accurate”, since the authors are basing this point on a single paper.

Response:

Thank you for asking the language to not state so objectively the accuracy of the GDS-15. We agree to modify it according to your suggestion. Indeed, this point is based on a single paper but not a single study. Fortunately, this systematic review and meta-analysis provided us a better lens to GDS-15 than if we just arbitrarily cited several papers, which unlikely covered the previous studies in a comprehensive way. The revision is tracked and commented in the manuscript, and also shown below.

Revision:

To screen depressive symptoms in older adults, the Geriatric Depression Scale (GDS) is one of the most commonly used questionnaires. A recent systematic review and meta-analysis found that the 15-item version (GDS-15) is the most accurate compared to the shorter or longer versions (Krishnamoorthy et al., 2020). 

Reference:

##### Krishnamoorthy, Y., Rajaa, S., and Rehman, T. (2020). Diagnostic accuracy of various forms of geriatric depression scale for screening of depression among older adults: Systematic review and meta-analysis. Arch Gerontol Geriatr 87, 104002. 10.1016/j.archger.2019.104002.

#### 2.2.2 #### I recommend removing the statement “questions asked later in the long-term”, and as suggested in my response to 2.1.4, I do not think response bias is a good motivation for this prediction model. I recommend removing this section.

Response:

We really appreciate this suggestion. The statement was removed and substituted with one focusing on the model as an EHR indicator. The revision is tracked and commented in the manuscript, and also shown below.

Revision:

By questionnaire-free variables, demographic and physical health data from routine visits can be utilized as an electronic health record (EHR) indicator to triage patients for a mental health follow-up by GDS-15. 

#### 2.2.4 #### I understand the authors’ point regarding the preference for machine learning. I recommend stating it explicitly upfront in the introduction, clarifying the word "insufficient" within the same sentence. For example: “[...] using logistic regression (LR) may be insufficient because its simplicity does not accurately reflect the complexity of real-world data [citation justifying this conclusion]. Machine learning [...]”.

Response:

Thank you for your suggestion. However, we disagree to explain the insufficiency due to its simplicity uncovering the real-world complexity. Such view is considerably disrespectful in another field which used regression algorithms in the majority of prediction studies, if not all. We tried to avoid dichotomy between LR and non-LR, and focus on problem-centric instead of method-centric solutions. In this study, as described in the main text, we tried to enlarge our hypothesis space of an accurate model of GDS-15 prediction, not limited to a probability theory. No revision is shown in the manuscript. 

################################################################################

######## 3 ######## Reviewer 3

################################################################################

I can see that authors have incorporated most of the changes as asked by the reviewers.

Response: 

Thank you for confirming our revision to the previous reviewer’s comments. Here below we described more revisions according to your comments.

###### 3.1 ###### Novelty of the proposed approach is either missing or the authors fail to express it in the manuscript. Novelty should be explained.

Response:

Thank you for pointing out this issue. We have already described the novelty, but we also believe it is failed to stand out in the manuscript. Since this is not a methodological paper, our novelty description focuses on the application novelty. To signify the novelty description, we added sub-sections in the Introduction. The novelty is described in sub-section “Previous works”. The revision is tracked and commented in the manuscript, and also shown below.

Revision:

Previous works

Almost all existing predictive models of depressive symptoms include questionnaire-based predictors, e.g., the Patient Health Questionnaire (PHQ), the Edinburgh Postnatal Depression Scale (EPDS), and the GDS (Smithson and Pignone, 2017). More-frequent identification of patients with depression (with an area under the receiver operating characteristic [ROC] curve [AUROC] of 0.700, 95% CI 0.629 to 0.771) still needed several questionnaire-based screening tools (Chekroud et al., 2016). They were a part of the Self-Reported Quick Inventory of Depressive Symptomatology (QIDS-SR) and Hamilton Depression Rating Scale (HAM-D). A previous study developed an extended predictD algorithm to predict major depression 12~24 months later based on the DSM-IV (AUROC 0.728, 95% CI 0.675 to 0.781; n=2670), but this also needs a subject to fill in the 12-Item Short Form (SF-12) for two of the predictors (King et al., 2013).

One study utilized a wearable device to predict GDS-15 and HAM-D results in older adults (AUROC 0.96, 95% CI 0.91 to 0.99; n=47) (Kim et al., 2019). Unfortunately, the sample size was small, and a wearable device might not be affordable for some older adults. However, no previous study developed a questionnaire-free method to predict depressive symptoms based on standard screening questionnaires in community-dwelling older adults.

References:

##### Chekroud, A.M., Zotti, R.J., Shehzad, Z., Gueorguieva, R., Johnson, M.K., Trivedi, M.H., Cannon, T.D., Krystal, J.H., and Corlett, P.R. (2016). Cross-trial prediction of treatment outcome in depression: a machine learning approach. Lancet Psychiatry 3, 243-250. 10.1016/s2215-0366(15)00471-x.

##### Kim, H., Lee, S., Lee, S., Hong, S., Kang, H., and Kim, N. (2019). Depression Prediction by Using Ecological Momentary Assessment, Actiwatch Data, and Machine Learning: Observational Study on Older Adults Living Alone. JMIR Mhealth Uhealth 7, e14149. 10.2196/14149.

##### King, M., Bottomley, C., Bellón-Saameño, J., Torres-Gonzalez, F., Svab, I., Rotar, D., Xavier, M., and Nazareth, I. (2013). Predicting onset of major depression in general practice attendees in Europe: extending the application of the predictD risk algorithm from 12 to 24 months. Psychol Med 43, 1929-1939. 10.1017/s0033291712002693.

##### Smithson, S., and Pignone, M.P. (2017). Screening Adults for Depression in Primary Care. Med Clin North Am 101, 807-821. 10.1016/j.mcna.2017.03.010.

###### 3.2 ###### Authors should explain the contributions made in the paper by adding a sub-section in the first section along with a sub-section on motivation.

Response:

Thank you for this suggestion. We have added several sub-sections in the first section, including a sub-section on “Motivation”. We also explained our contribution in sub-sections “Previous works” and “Intuition”. The revision is tracked and commented in the manuscript, and the description on contribution is also shown below.

Revision:

However, no previous study developed a questionnaire-free method to predict depressive symptoms based on standard screening questionnaires in community-dwelling older adults. 

Intuition 

Later-life (aged 60+) depression is associated with several factors, and their assessments can utilize routine databases at the first visit of a subject to a healthcare facility.

###### 3.3 ###### Paper requires a flowchart which can show the flow and step by step approach of the proposed methodology in tackling the undertaken problem.

Response:

Thank you for this suggestion. We have added the flowchart as Figure 1. The revision is tracked and commented in the manuscript, and also shown below.

Figure 1. Flowchart of the proposed methodology.

###### 3.4 ###### Formatting mistakes are there; such as random forest [RF] , here it should be written as random forest (RF), Similar errors should be removed.

Response:

Thank you for your suggestion. Previously, we used square brackets for bracket nesting to avoid using parentheses inside parentheses, in order to keep formatting consistently in US English. We have rechecked and did not find if we used square brackets outside parentheses. No revision is made for this comment.

---

## [Editor Report · Decision Letter 2]

27 Dec 2022

Questionnaire-free machine-learning method to predict depressive symptoms among community-dwelling older adults

PONE-D-22-10670R2

Dear Dr. Chuang,

We’re pleased to inform you that your manuscript has been judged scientifically suitable for publication and will be formally accepted for publication once it meets all outstanding technical requirements.

Kind regards,

Tarik A. Rashid, PhD

Academic Editor

PLOS ONE

---

## [Editor Report · Acceptance letter]

3 Jan 2023

PONE-D-22-10670R2 

Questionnaire-free machine-learning method to predict depressive symptoms among community-dwelling older adults 

Dear Dr. Chuang:

I'm pleased to inform you that your manuscript has been deemed suitable for publication in PLOS ONE. Congratulations! Your manuscript is now with our production department. 

Kind regards, 

on behalf of

Dr. Tarik A. Rashid 

Academic Editor

PLOS ONE